# Towards Foundational Models for Dynamical System Reconstruction: Hierarchical Meta-Learning via Mixture of Experts

## Abstract

As foundational models reshape scientific discovery, a bottleneck persists in dynamical system reconstruction (DSR): the ability to learn across system hierarchies. Many meta-learning approaches have been applied successfully to single systems, but falter when confronted with sparse, loosely related datasets requiring multiple hierarchies to be learned. Mixture of Experts (MoE) offers a natural paradigm to address these challenges. Despite their potential, we demonstrate that naive MoEs are inadequate for the nuanced demands of hierarchical DSR, largely due to their gradient descent-based gating update mechanism which leads to slow updates and conflicted routing during training. To overcome this limitation, we introduce MixER: Mixture of Expert Reconstructors, a novel sparse top-1 MoE layer employing a custom gating update algorithm based on $K$-means and least squares. Extensive experiments validate MixER's capabilities, demonstrating efficient training and scalability to systems of up to ten parametric ordinary differential equations. However, our layer underperforms state-of-the-art meta-learners in high-data regimes, particularly when each expert is constrained to process only a fraction of a dataset composed of highly related data points. Further analysis with synthetic and neuroscientific time series suggests that the quality of the contextual representations generated by MixER is closely linked to the presence of hierarchical structure in the data.

## 1. Introduction

The emergence of foundational models in language and vision has catalyzed an accelerated pursuit of analogous models for scientific discovery (Subramanian et al., 2024; Herde et al., 2024). Unlike traditional data modalities, scientific data presents unique challenges due to its inherent complexity and scarcity. This challenge has motivated the development of sophisticated dynamical system reconstruction (DSR) models capable of robust generalization across varying domains—with each variation constituting an *environment*. However, the effectiveness of these systems in low-data scenarios hinges on substantial relatedness among environments, raising fundamental questions about learning across **families** of loosely connected environments (see Figure 1).

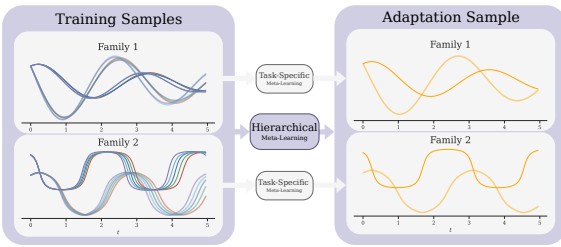

*Figure 1.* Task-specific and hierarchical meta-learning. Each family comprises a set of environments defined by the same ordinary differential equation (ODE). Within a family, parameters of the underlying ODE are varied, producing dynamics that are similar but unique. Task-specific meta-training focuses on adaptation within a family, while hierarchical meta-learning enables simultaneous training across families, followed by adaptation to any of them.

Usual approaches to data-driven generalizable DSR (Göring et al., 2024) predominantly rely on Expected Risk Minimization (ERM) (Sagawa et al., 2020; Brandstetter et al., 2022), assuming abundant environment-specific samples to capture the full spectrum of observable dynamics. This assumption proves problematic in data-scarce domains like clinical or neuroscience applications (Brenner et al., 2024). While multitask learning (Yin et al., 2021) has emerged as a popular alternative, it lacks robust adaptation mechanisms for novel scenarios. Recent advances in **meta-learning** (Wang et al., 2021; Finn et al., 2017) have demonstrated remarkable success by explicitly incorporating adaptation capabilities into the training process. *Contextual* meta-learning (Nzoyem et al., 2024) achieves this through a strategic separation

[1]Anonymous Institution, Anonymous City, Anonymous Region, Anonymous Country. Correspondence to: Anonymous Author <anon.email@domain.com>.

Preliminary work. Under review by the International Conference on Machine Learning (ICML). Do not distribute.

of parameters into environment-agnostic components and compact context vectors amenable to environment-specific fine-tuning via gradient descent. Current state-of-the-art approaches are categorized in two primary paradigms: *hyper-network*-based methods (Kirchmeyer et al., 2022; Brenner et al., 2024; Roeder et al., 2019; Koupaï et al., 2024) that condition environment-specific weights on context, and *concatenation*-based alternatives (Nzoyem et al., 2025; Zintgraf et al., 2019) that directly feed the context to the dynamics-generating model. Despite their strong potential, meta-learning approaches exhibit limitations when confronted with environments that have minimal or no similarities, which is the case when the time series are not governed nor driven by the same underlying differential equations.

Drawing inspiration from recent breakthroughs in large language modeling (Liang et al., 2024; Jiang et al., 2024; Dai et al., 2024; Abnar et al., 2025), we investigate the potential of augmenting existing meta-learners with sparse **mixture of experts** (MoEs) (Jacobs et al., 1991; Shazeer et al., 2017b) for generalizable DSR. Despite inherent routing challenges that constrain their applications to DSR, MoEs offer a natural framework for learning across families of arbitrarily related environments. We claim that strategic combination of contextual meta-learners enables simultaneous reconstruction across all families while preserving rapid adaptation capabilities, obviating the need for manual dataset partitioning prior to meta-learning on each subset.

Specifically, we introduce a novel routing mechanism for MoEs that facilitates learning across dynamical systems and potentially unrelated families thereof, while maintaining adaptability within individual environments. Our approach achieves this through two key innovations: $(i)$ grounding the routing mechanism in contextual information rather than state input vectors, and $(ii)$ implementing an explicit routing protocol combining $K$-means clustering with least squares optimization for precise expert-cluster pairing.

After establishing the formal problem structure and highlighting the imperative for hierarchical meta-learning in Section 2, we present our MixER methodology and its core optimization components in Section 3. Section 4 demonstrates our main findings in few-shot learning and time series classification, benchmarked against current state-of-the-art approaches. We summarize our contributions as follows:

1. We identify a fundamental limitation of gradient-descent when routing contextual information to DSR models, which slows down expert specialization when training MoEs.

2. We propose an effective unsupervised routing mechanism for MoEs to collectively learn dynamical systems with various degrees of relatedness.

3. We provide experimental evidence of the breadth of applicability of our method on two and ten families of ordinary differential equations, several classical DSR benchmarks, and synthetic time-series data.

4. We demonstrate the limited contextual representations resulting from MixER when the data is ambiguous and closely related, as is the case with neuroscientific epileptic data.

## 2. Problem Description & Motivation

This work addresses the fundamental challenge of learning collections of dynamical systems from data with limited knowledge of their interrelationships. We demonstrate the limitations of conventional meta-learning approaches through two ordinary differential equations (ODEs), and we propose MixER as a solution to these limitations.

### 2.1. Hierarchical Dynamical System Reconstruction

In this section, we present the setting of hierarchical DSR. For notational clarity, we employ consistent shorthand throughout the paper. The notation $[T]$ denotes the discrete set $\{1, \ldots, T\}$ for any positive integer $T$. Environment-related indices (e.g., $e$, $i$, and $f$ as seen below) are denoted by *superscripts*, while other indices such as time $t$ and expert count $m$ use *subscripts*.

The reconstruction of families of dynamical systems requires a novel framework for handling multi-level temporal data. In our framework, each datum consists of a (multi-variate) time series $\{x_t\}_{t \in [T]} \in \mathbb{R}^{T \times d}$ of length $T > 0$ and dimension $d \geq 1$, representing either simulated trajectories or observed process measurements. These data points may present shared knowledge, such as repeated clinical measurements from a patient (Brenner et al., 2024) or varying initial conditions of the same physical system, referred to as "environments". The complete dataset comprises $E \geq 1$ environments $\{x_t^{e,i}\}_{i \in [I]}^{e \in [E]}$, where $I \geq 1$ represents the distinct time series count in environment $e$. When environments exhibit higher-order relationships, the dataset extends to $F \geq 1$ *families*, denoted as $\{x_t^{f,e,i}\}^{f \in [F]}$ (see Figure 1).

Importantly, we make no assumptions about inter-family relationships, which may range from loose to intricate connections. For this reason, the training data is presented as $\mathcal{D}_{\text{tr}} \triangleq \{x_t^{e,i}\}^{e \in [E]}$, with unsupervised environment clustering into families occurring during learning. In cases without repeated measurements, each time series $i$ constitutes its own environment. This framework enables the development of foundational models capable of processing heterogeneous data while generalizing conventional dynamical system reconstruction approaches.

Learning on our datasets can be viewed in two levels: conventional (or flat), and hierarchical DSR models.

**Flat DSR Models** The base level formulates dynamical system reconstruction as a supervised learning problem (Göring et al., 2024; Kramer et al., 2021; Yin et al., 2021). The primary objective is learning a flow mapping $G_\theta$ that transforms latent representation $z_t$ across time steps:

$$z_t = G_\theta(z_{t-1}, x_{t-1}), \qquad (1)$$

where $x_{t-1}$ represents an optional ground truth teacher-forcing signal and $\theta$ denotes learnable parameters. We note, however, that $x_{t-1}$ is *not* used during inference as the system is rolled out auto-regressively. This formulation describes a *sequence-to-sequence* learning problem (Brenner et al., 2024; Kidger et al., 2020; Gu & Dao, 2023).

In scenarios without teacher forcing (Yin et al., 2021; Kirchmeyer et al., 2022), the problem transforms into a *state-to-sequence* or *initial value problem* (IVP):

$$\frac{dz_t}{dt} = G_\theta(z_t), \qquad \forall t \in [0, T], \qquad (2)$$

or equivalently:

$$z_t = z_0 + \int_0^t G_\theta(z_\tau)\, d\tau, \qquad \forall t \in [0, T]. \qquad (3)$$

This approach underlies Neural ODEs (Chen et al., 2018; Rackauckas et al., 2020; Haber & Ruthotto, 2017; Weinan, 2017), which have become invaluable in generative modeling (Lipman et al., 2022; Liu et al., 2023) and engineering applications (Kochkov et al., 2024; Shen et al., 2023).

**Hierarchical DSR Models** Environment-aware models introduce a context vector $\xi$ that modulates model behavior. We consider two conditioning approaches: *hypernetwork*-based conditioning (Kirchmeyer et al., 2022; Brenner et al., 2024), where a secondary network $H_\theta$ generates environment-specific weights:

$$z_t = G_{\theta^e}(z_{t-1}, x_{t-1}^e), \quad \text{with } \theta^e = H_\theta(\xi^e), \qquad (4)$$

and *concatenation*-based conditioning (Nzoyem et al., 2025; Zintgraf et al., 2019), where the context is directly fed to the flow map:

$$z_t = G_\theta(z_{t-1}, x_{t-1}^e, \xi^e). \qquad (5)$$

For convenience, both approaches can be denoted as

$$z_t = G_{\theta,\xi^e}(z_{t-1}, x_{t-1}^e). \qquad (6)$$

Current hierarchical DSR models, such as Equation (6), struggle with complex data relationships (e.g. families of unrelated environments), raising the critical question:

*What is the optimal way to cluster environments so that existing contextual meta-learning approaches can utilize them effectively?*

Before presenting our solution to this question, we underline the central limitation of contextual meta-learning with a simple example. We show that without appropriate clustering, these methods become ineffective when confronted with datasets of loosely related families.

## 2.2. Motivating Example

The limitations of task-specific meta-learners and naive MoE become evident in the initial value problem (IVP) setting of Figure 1. Our test case involves simultaneous learning of two 2-dimensional ODEs proposed by d'Ascoli et al. (2024). The dataset comprises two families, with the goal of reconstructing $I = 32$ test-time trajectories across $E = 10$ total environments from both families (see ODEBench-2 in Table 1). We evaluate three state-of-the-art meta-learners—Neural Context Flow (NCF) (Nzoyem et al., 2025), CoDA (Kirchmeyer et al., 2022), and GEPS (Koupaï et al., 2024)—within a top-1 MoE framework.

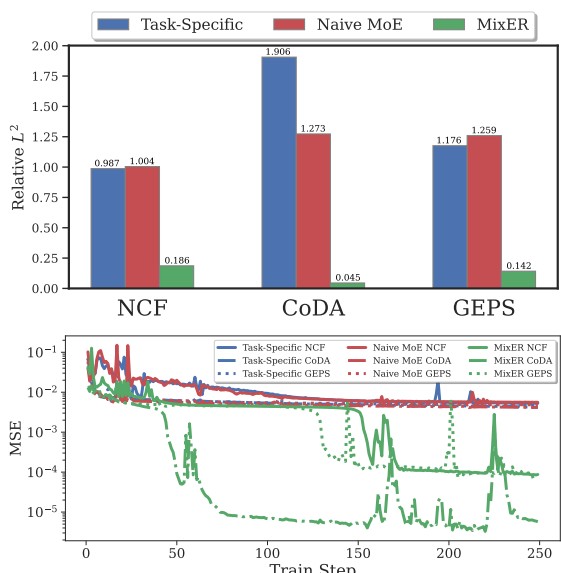

*Figure 2.* Limitation of task-specific meta-learning and vanilla MoE on the two families of ODEs from Figure 1. Strategically increasing the capacity of the network with MixER and its special routing algorithm results in a successful model. (Top) Relative $L^2$ error on test set; (Bottom) Validation MSE losses during training.

Figure 2 demonstrates that single task-specific meta-learners cannot capture the inherent data complexity. Furthermore, a naive MoE implementation with gradient-based gating updates (Shazeer et al., 2017a) routes all contexts to a single expert (Figure 8), yielding suboptimal performance. The validation losses reveal that once suitable gating weights are found and family-expert pairings established, our proposed solution (in green) dramatically improves performance starting around training step 40 for CoDA or 150 for NCF.

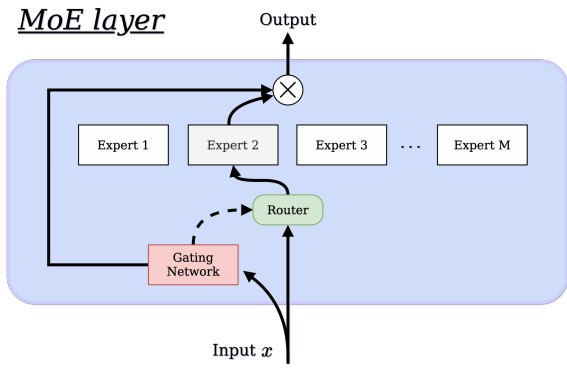

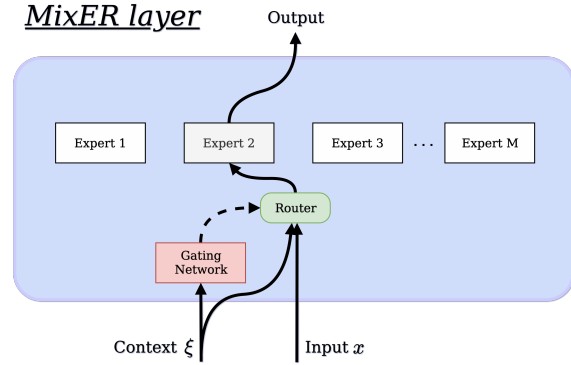

*Figure 3.* Illustration of vanilla MoE and our proposed MixER layer. **(Left)** Vanilla MoE setting where a single input $x$ is passed through a gating network whose outputs enable the router to assign computation to a specific expert (Chen et al., 2022). **(Right)** Our sparse MixER layer requires a context vector $\xi$ alongside the input $x$. The gating network computes expert affinities based on this context vector. Contrary to MoE, the MixER layer disregards the softmax-weighted output aggregation.

Top-1 MoE's fundamental advantage lies in its reduced active parameter count which saves computation during inference (Jiang et al., 2024; Fedus et al., 2022). The following section details our improved routing mechanism which leverages this top-1 sparsity structure.

## 3. Mixture of Expert Reconstructors

We present a novel MoE layer that leverages $K$-means clustering and least squares to minimize routing conflicts across DSR models in its layer.

The MixER layer, depicted in Figure 3, fundamentally differs from vanilla MoE layers in two aspects. First, MixER incorporates an environment-specific context vector $\xi$ as additional input for computing gating weights, addressing the limitation that pointwise state input $x_t$ alone cannot fully characterize temporal behavior. Second, MixER employs a top-1 MoE architecture and eliminates the need for softmax weighting of expert outputs. This design choice enables experts to function independently outside the layer, a critical feature for our gating network update methodology.

### 3.1. Optimization Procedure

Our training pipeline optimizes both environment-specific parameters $\Xi \triangleq \{\xi^e\}^{e \in [E]}$ and shared parameters $\Theta \triangleq \{\theta_m\}_{m \in [M]}$, where $M$ denotes the number of experts. The optimization minimizes the aggregate MSE loss:

$$\mathcal{L}(\Theta, \Xi, \mathcal{D}_{\text{tr}}) \triangleq \frac{1}{E \times I \times T} \sum_{e=1}^{E} \sum_{i=1}^{I} \sum_{t=1}^{T} \|\hat{x}_t^{e,i} - x_t^{e,i}\|_2^2, \quad (7)$$

where $\hat{x}$ represents the reconstructed trajectory. We implement *proximal* alternating minimization (Li et al., 2019), chosen for its proven convergence to second-order optimal solutions (Li et al., 2019; Nzoyem et al., 2025). Notably,

our framework eliminates the need for importance or load-balancing terms in the loss function (Shazeer et al., 2017b).

The gating network $W$ updates occur independently of $\Theta$, as motivated in Section 2.2. Our implementation applies gating updates after each (or several) gradient update of either $\Theta$ or $\Xi$. During adaptation to novel environments, only context vectors undergo optimization via gradient descent, while $W$ and $\Theta$ remain fixed.

To address scale variations across trajectory families, we employ small batches of closely related environments (determined by $L^1$ norm between context vectors) for stochastic updates of $\Theta$ and $\Xi$. For validation and model selection in these scenarios, we utilize the relative $L^2$ loss defined in Equation (9).

### 3.2. Gating Network Update

The gating network transforms a context $\xi^e$ into $M$ logits $g^e \triangleq \{g_m^e\}_{m \in [M]}$, where the maximum value identifies the optimal expert for environment $e$. We implement a linear mapping[1]:

$$g^e = \xi^e W, \qquad \forall\, e \in \{1, \dots, E\} \quad (8)$$

optimized through least squares (see Algorithm 1), with labels $Y$ (proxies for $g^e$) derived from $K$-means clustering using Lloyd's algorithm (Lloyd, 1982) (see Algorithm 2).

The update procedure in Algorithm 1 comprises four key stages: **(1)** $K$-means clustering (line 6); **(2)** per-expert per-environment loss computation (lines 7 and 8); **(3)** expert-cluster pairings (lines 9 to 18); and **(4)** least-squares optimization (lines 19 to 24). These stages are visualized in Figure 4.

---

[1] In practice, we note that $W$ contains a bias term omitted here for conciseness.

**Algorithm 1** Gating Network Update

1: **Require:** $\Theta := \{\theta_m\}_{m\in[M]}$ mixture of $M$ experts
2: $\Xi := \{\xi^e\}^{e\in[E\times F]}$ context vectors
3: $\bar{\bar{\Xi}} := \{\bar{\xi}_m\}_{m\in[M]}$ centroid initialization
4: $\mathcal{D}_{\text{tr}} \triangleq \{\mathcal{D}_{\text{tr}}^e\}^{e\in[E]}$ training data
5: $\sigma > 0$ noise standard deviation
6: $C, \bar{\Xi} \leftarrow K\text{-Means}(\Xi, \bar{\bar{\Xi}})$  ▷ see Algorithm 2

7: $\ell_{m,e} = \mathcal{L}(\theta_m, \xi^e, \mathcal{D}_{\text{tr}}^e)$   $\forall m \in [M], \forall e \in [E]$
8: $\bar{\ell}_{\cdot,c} = \text{Median}\{\ell_{\cdot,e} : e \in C_c\}$   $\forall c \in [M]$

9: Initialize $\mathcal{S} \leftarrow \emptyset$  ▷ Selected experts
10: **for** $c \in [M]$ **do**
11: SortedList $\leftarrow \text{argsort}(\bar{\ell}_{\cdot,c})$
12: $m \leftarrow \text{SortedList}_1$
13: **while** $m \in \mathcal{S}$ **do**
14: SortedList $\leftarrow \text{SortedList}_{2:\text{length(SortedList)}}$
15: $m \leftarrow \text{SortedList}_1$
16: **end while**
17: $\mathcal{S} \leftarrow \mathcal{S} \cup \{m\}$
18: **end for**

19: $Y \leftarrow \mathbf{0}_{E\times M}$  ▷ Least squares proxy labels
20: **for** $c \in [M]$ **do**
21: $Y_{C_c} \leftarrow \text{OneHotEncode}(\mathcal{S}_c, M)$
22: **end for**
23: $X \leftarrow \Xi + \mathcal{N}(0, \sigma)$  ▷ Add noise to context
24: $W \leftarrow \text{LeastSquares}(X, Y)$

25: **Return** $W, \bar{\Xi}$

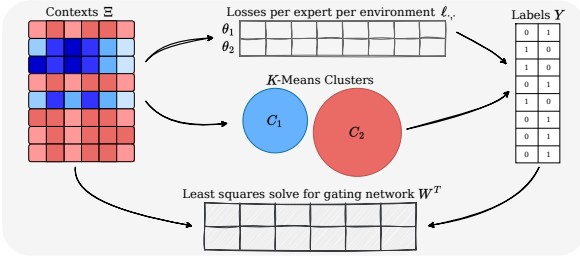

*Figure 4.* Illustration of the main stages of our context-based gating update algorithm, with all tensors in compatible shapes.

Our implementation incorporates two crucial optimizations. First, we mitigate $K$-means sensitivity to initial conditions by reusing centroids from previous gating updates (line 6), achieving convergence typically within two iterations. Second, we introduce controlled noise to $\Xi$ before least squares computation, enhancing robustness against suboptimal configurations and preventing instability during early training when context values cluster near their zero initialization.

# 4. Experiments

We evaluate our approach through comprehensive experiments on both loosely and closely related dynamical systems, synthetic and real-world datasets. Our analysis encompasses datasets of varying complexity, baseline comparisons, and detailed performance assessments.

## 4.1. One-Shot Learning on Loosely Related Families

Meta-learning across families of dynamical systems demonstrates the potential of our approach. Using the ODEBench dataset (d'Ascoli et al., 2024), we analyze 10 distinct ODE families, each containing multiple environments generated by parameter variations (Table 1). The experimental setup consists of 4 meta-training trajectories per environment, with 32 additional trajectories reserved for evaluation. One-shot adaptation is evaluated by fine-tuning context vectors on a single trajectory, repeated across 4 adaptation environments per family. Further data generation details are available in Appendix B.

*Table 1.* Number of training families and environments extracted from the ODEBench dataset (d'Ascoli et al., 2024).

|  | # Families | # Env. Per Fam. | # Total Envs. |
|---|---|---|---|
| **ODEBench-2** | 2 | 5 | 10 |
| **ODEBench-10A** | 10 | 5 | 50 |
| **ODEBench-10B** | 10 | 16 | 160 |

As in Section 2.2, we consider three leading adaptation rules: NCF (Nzoyem et al., 2025), CoDA (Kirchmeyer et al., 2022), and GEPS (Koupaï et al., 2024). We wish to know whether our approach boosts the performance of these baselines on such loosely connected data. All adaptation rules utilize the same MLP (Haykin, 1994) as the root (or main) network. Our MixER implementations employ context vectors of dimension $d_\xi = 40$, evenly distributed among expert meta-learners (implementation details in Appendix A).

*Table 2.* Training and adaptation relative MSEs ($\downarrow$) on the ODEBench-10A dataset, across 3 runs with different seeds. MixER-$M$ means $M$ experts are present in the layer. The † indicates the naive MoE with the gate updated via gradient descent. The best along the columns is reported in **bold**.

|  | NCF | | CoDA | | GEPS | |
|---|---|---|---|---|---|---|
|  | TRAIN | ADAPT | TRAIN | ADAPT | TRAIN | ADAPT |
| MIXER-1 | 2.05±0.12 | **1.80±0.28** | 0.98±0.08 | 6.91±1.25 | 2.61±0.1 | 2.20±0.21 |
| MIXER-10† | 1.53±0.34 | 5.28±1.04 | 0.76±0.07 | **4.25±0.15** | **0.58±0.04** | **1.16±0.09** |
| MIXER-10 | **1.05±0.09** | 2.38±0.23 | **0.47±0.06** | 15.9±4.2 | 1.01±0.05 | 1.29±0.08 |

Results on ODEBench-10A (Table 2) reveal that MixER enhances performance across all contextual meta-learning backbones on the training evaluation sets. However, adaptation performance varies significantly, with GEPS main-

taining consistency while NCF and CoDA's performance degrades. The limited environment count in ODEBench-10A constrains overall performance, motivating our evaluation on the more comprehensive ODEBench-10B dataset.

Table 3. Training and adaptation metrics on ODEBench-10B. (Top) Relative MSE ($\downarrow$); (Bottom) Proportion of environments (%) with relative MSE below the threshold of $\varepsilon = 0.1$ ($\uparrow$).

| | NCF | | CoDA | | GEPS | |
|---|---|---|---|---|---|---|
| | Train | Adapt | Train | Adapt | Train | Adapt |
| MixER-1 | **0.12±0.01** | 3.20±0.28 | **0.07±0.01** | **0.34±0.05** | 0.21±0.1 | 1.24±0.07 |
| MixER-10† | 0.29±0.02 | 2.53±0.20 | 0.15±0.03 | 0.72±0.08 | 0.13±0.4 | **0.49±0.02** |
| MixER-10 | 0.22±0.60 | 1.43±0.23 | 0.10±0.02 | 14.8±4.2 | **0.06±0.01** | 1.43±0.02 |
| MixER-20 | 0.38±0.02 | **0.54±0.02** | 0.12±0.04 | 0.38±0.02 | 0.17±0.03 | 0.92±0.10 |

| | NCF | | CoDA | | GEPS | |
|---|---|---|---|---|---|---|
| | Train | Adapt | Train | Adapt | Train | Adapt |
| MixER-1 | **66.9±2.1** | 40.0±2.3 | **85.6±2.7** | 50.0±2.5 | 64.4±3.9 | 37.5±3.5 |
| MixER-10† | 36.9±6.4 | 30.0±3.5 | 71.9±1.6 | 42.5±0.8 | 71.2±5.4 | **50.0±1.4** |
| MixER-10 | 55.6±4.6 | 2.5±2.8 | 80.0±2.5 | 25.0±2.8 | **85.6±5.4** | 32.5±3.0 |
| MixER-20 | 37.5±2.4 | 20.0±4.2 | 60.6±8.0 | 27.5±1.0 | 55.6±3.2 | 35.0±2.5 |

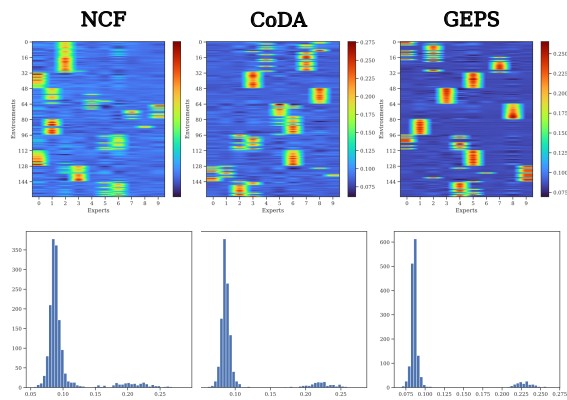

Figure 5. Gating weights on ODEBench-10B, at the end of training with MixER-10. (Top) Gating heatmap. (Bottom) Histogram across all 160 environments.

Analysis of ODEBench-10B (Table 3) shows that MixER-10 underperforms compared to MixER-10† (naive MoE) and MixER-1 (a single meta-learner). Performance analysis using relative $L^2$ thresholds (defined in Equation (10)) globally indicates diminished benefits from using MixER-10. However, GEPS exhibits remarkable robustness, showing consistent improvement with increased expert count in both training and adaptation scenarios, even with gradient-based gating updates. As expected, visualization of gating values (Figure 5) reveals that enhanced performance correlates with improved environment-to-expert routing in groupings of 16 across all 160 environments.

### 4.2. Generalization on Classical DSR Datasets

Classical DSR benchmarks reveal the breadth of applicability of our approach. We evaluate three datasets of closely related environments: ($i$) Lotka-Volterra (**LV**), a 2-dimensional ODE modeling species evolution in closed ecosystems (Yin et al., 2021); ($ii$) Glycolytic Oscillator (**GO**), a model of yeast glycolysis (Kirchmeyer et al., 2022); and ($iii$) Sel'kov Model (**SM**), a more complex 2-dimensional ODE for glycolysis that exhibits a Hopf bifurcation (Nzoyem et al., 2025).

We consider the same NCF, CoDA, and GEPS backbones as above. Additionally, we consider CAVIA[2] (Zintgraf et al., 2019). MixER employs three experts across all experiments, with parameter counts matched to baselines for fair comparison. Context vector dimensions vary by backbone: NCF uses $d_\xi = 512$, while CoDA and GEPS use $d_\xi = 2$, reflecting underlying physical parameter variations. Additional hyperparameters are documented in Appendix C.

Results presented in Table 4 demonstrate that while all methods successfully approximate the IVP vector fields, MixER underperforms relative to its baseline meta-learners. Clustering and routing analysis (Figure 9) shows that MixER logically partitions datasets into three subsets, but this partitioning limits each expert meta-learner's exposure to the full dataset, potentially explaining the performance degradation despite clear cross-environment commonalities.

### 4.3. Feature Interpretability and Downstream Clustering

A major benefit of contextual meta-learning is in its byproduct context features, which can be used for downstream tasks. To test the interpretability of these features, we consider two time series classification datasets. First, the Synthetic Control Chart Time Series (SCCTS) (Alcock, 1999) is a collection of 600 time series[5] across six classes: A. Normal, B. Cyclic, C. Increasing trend, D. Decreasing trend, E. Upward shift, and F. Downward shift. The traditional $K$-means typically struggles to separate these classes due to the similarities among the pairs A/B, C/D, and E/F. We expect the grouping (600 environments $\rightarrow$ 6 classes $\rightarrow$ 3 families) to be suitable for hierarchical models. As such, we train a MixER with 3 expert meta-learners in a completely *unsupervised* manner.

Second, the Epilepsy2 dataset (Andrzejak et al., 2001) is a large collection of real-world neuroscientific EEG data with noisy labels indicating whether a subject is healthy (0) or experiencing a seizure (1). The 80 unshuffled training samples are labeled as follows [0-30): 1, [30-60): 0, [60-70): 1, and [70-80): 0. For this dataset, our families are the two underlying classes. We emphasize that SoTA meth-

---

[2]We did not augment CAVIA within our MixER layer due to its second-order optimization algorithm.

[4]For integration within the MixER layer, we performed custom reimplementation of the backbones as explained in Appendix C.2.

[5]Each time series constitutes its own environment, i.e., $I = 1$.

*Table 4.* In-Domain (InD) and Out-of-Distribution (OoD) test MSEs (↓) for the LV, GO, and SM problems. The star indicates runs using the reference implementations[4]. Results for CAVIA, CoDA* and NCF* are reported from (Nzoyem et al., 2025). The best is reported in **bold**. The best of the three MixERs is shaded in grey . The #PARAMS columns indicate the active parameter counts.

| | **LV** ($\times 10^{-5}$) | | | **GO** ($\times 10^{-4}$) | | | **SM** ($\times 10^{-3}$) | | |
| --- | --- | --- | --- | --- | --- | --- | --- | --- | --- |
| | #PARAMS | InD | OoD | #PARAMS | InD | OoD | #PARAMS | InD | OoD |
| CAVIA | 305246 | 91.0±63.6 | 120.1±28.3 | 130711 | 64.0±14.1 | 463.4±84.9 | 50486 | 979.1±141.2 | 859.1±70.7 |
| CoDA* | 305793 | **1.40±0.13** | 2.19±0.78 | 135390 | 5.06±0.81 | 4.22±4.21 | 50547 | 156.0±40.52 | 8.28±0.29 |
| NCF* | 308240 | 1.68±0.32 | **1.99±0.31** | 131149 | **3.33±0.14** | **2.83±0.23** | 50000 | **6.42±0.41** | **2.03±0.12** |
| MIXER-NCF | 307245 | 3.70±0.4 | 4.45±0.3 | 130535 | 73.5±21.1 | 141.5±82.8 | 50387 | 32.3±4.2 | 64.2±1.5 |
| MIXER-CoDA | 307995 | 4.00±0.01 | 53.5±0.4 | 132137 | 42.0±18.9 | 49.3±25.1 | 51995 | 32.8±3.9 | 317.2±6.0 |
| MIXER-GEPS | 305112 | 14.8±0.7 | 82.4±0.9 | 131747 | 22.3±23.2 | 259.7±45.0 | 51312 | 27.6±5.8 | 46.3±2.7 |

ods do not face difficulties classifying this data, whereas naive $K$-means consistently struggles. For both SCCTS and Epilepsy2 datasets, the backbone meta-learner we use is the hier-shPLRNN (Brenner et al., 2024) based on the generalized teacher forcing approach (Hess et al., 2023). We fix its mixing coefficient $\alpha = 0.5$, the hidden layer's width to 16, and we use a linear hypernetwork to generate weights based on context vectors of size $d_\xi = 10$.

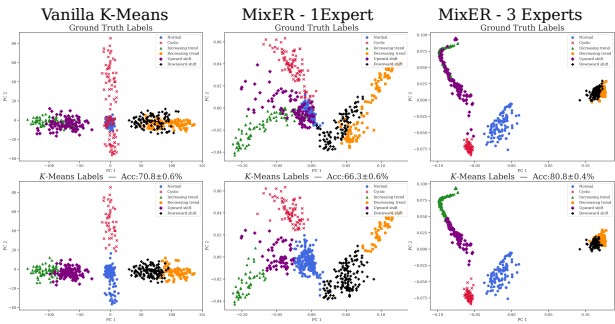

*Figure 6.* PCA clusters formed when training a MixER on the SCCTS dataset. (Top) Coloring using the ground truth labels; (Bottom) Coloring using labels from a $K$-means algorithm, with its means initialized at the ground truth means.

SCCTS results (Figure 6) demonstrate improved class separation with three experts, effectively grouping similar classes (A/B, C/D, E/F) and routing them to the same expert. This configuration unambiguously outperforms both single-expert and vanilla $K$-means approaches in qualitative and quantitative metrics.

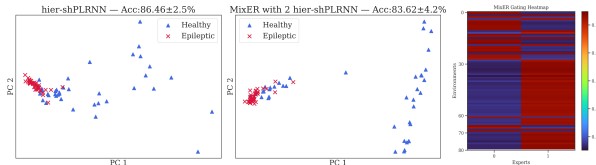

*Figure 7.* PCA clusters on the Epilepsy2 datasets, using the hier-shPLRNN meta-learning backbone. Accuracy scores are obtained on the testing contexts upon training a logistic regression classifier.

Conversely, Epilepsy2 results (Figure 7) show degraded performance with MixER. While context routing roughly aligns with class boundaries, the clusters lack clear separation, with epileptic subjects split between experts while healthy subjects route exclusively to the second expert. This routing pattern persists in test data, challenging downstream classification via logistic regression (Cox, 1958). The classification performance degradation likely stems from the dataset's inherent noise, as noted by Brenner et al. (2024). Indeed, such close proximity of time series prevents clean discrimination and routing during training. These results highlight MixER's limitations with ambiguous, highly related environments.

## 5. Related Work

We review the emerging field of dynamical system reconstruction (DSR) and its intersection with meta-learning for multi-environment generalization. We cover learning generalizable DSRs and their extension to foundational models.

**Multi-Environment Learning** The challenge of multi-environment learning has received substantial attention in the machine learning community. Contemporary multi-domain training approaches extend the traditional Empirical Risk Minimization (ERM) framework through Invariant Risk Minimization (IRM) (Arjovsky et al., 2019) and Distributionally Robust Optimization (DRO) (Ben-Tal et al., 2013; Sagawa et al., 2020; Krueger et al., 2021), which optimize models to minimize worst-case performance across potential test distributions. For optimal reconstruction of ODEs, PDEs, and differential equation-driven time series, several models incorporate physical parameters as model inputs (Brandstetter et al., 2022; Takamoto et al., 2023). This approach assumes that exposure to training physical parameters enables models to learn the underlying parameter distribution and its relationship to system dynamics. However, these physical parameters are often sparse or unobservable, necessitating the learning of data-driven proxies through multitask learning (MTL) (Caruana, 1997) and meta-learning (Hospedales et al., 2021) approaches for

DSRs. While MTL methods typically adapt components of a generalist model across training environments (Yin et al., 2021), they often lack the rapid adaptation capabilities of their meta-learning counterparts when confronted with out-of-distribution scenarios.

**Generalization to New Environments** Meta-learning, embodied by *adaptive conditioning* (Serrano et al., 2024) in the DSR community, represents the primary framework for generalization. Rather than conducting complete model fine-tuning for each new environment (Subramanian et al., 2024; Herde et al., 2024), this approach implements training with rapid adaptation in mind. *Contextual* meta-learning partitions learnable parameters into environment-agnostic and environment-specific components. These contexts serve diverse purposes: (1) Encoder-based methods (Garnelo et al., 2018; Wang et al., 2022) employ dedicated networks for context prediction, though they tend to overfit on training environments (Kirchmeyer et al., 2022). (2) Hypernetwork-based approaches (Kirchmeyer et al., 2022; Brenner et al., 2024; Blanke & Lelarge, 2024) learn transformations from context to model parameters. GEPS (Koupaï et al., 2024), through its LoRA-inspired adaptation rule (Hu et al., 2021), enhances these methods for large-scale applications. (3) Concatenation-based conditioning strategies (Zintgraf et al., 2019; Nzoyem et al., 2025) incorporate context as direct input to the model. While these frameworks demonstrate considerable efficacy, none directly addresses learning across families of arbitrarily related environments.

**Learning in Families of Environments** Clustering before training, followed by task-specific meta-learning (Nzoyem et al., 2025; Kirchmeyer et al., 2022; Koupaï et al., 2024; Brenner et al., 2024) would constrain the adaptability of our models. The challenge of simultaneous learning across arbitrarily related families remains largely unexplored, particularly in the context of Mixture of Experts (MoE) (Jacobs et al., 1991). MoE is a powerful paradigm, as Chen et al. (2022) demonstrate that certain tasks fundamentally require expert mixtures rather than single experts. Most relevant to our context is the variational inference approach of (Roeder et al., 2019; Davidian & Giltinan, 2003) which infers Neural ODE (Chen et al., 2018) parameters across well-defined hierarchies. The language modeling community provides compelling demonstrations of MoE efficacy (Shazeer et al., 2017b). Sparse MoEs enable expert MLPs to encode domain-specific knowledge (Dai et al., 2024; Jiang et al., 2024; Guo et al., 2025), while some MoE variants address catastrophic forgetting (He, 2024). Drawing inspiration from "switch routing" (Fedus et al., 2022), our work dedicates single experts to individual families during training.

**Foundational Scientific Models** Current foundational scientific models remain domain-specific, as exemplified in climate modeling (Nguyen et al., 2023; Bodnar et al., 2024) where abundant data sources maintain relative homogeneity. Kochkov et al. (2024) achieves generalization through the hybridization of principled atmospheric models with neural networks. While PINNs (Cuomo et al., 2022) underpin numerous powerful systems, they demand substantial data and domain expertise (Nzoyem et al., 2023). Our approach diverges by discovering physics from data without prior physical knowledge while maintaining adaptability. Although domain-agnostic models are emerging (Subramanian et al., 2024; Herde et al., 2024), they typically require resource-intensive pre-training and fine-tuning. To the best of our knowledge, our work represents the first DSR approach targeting such broad generalization through rapid adaptation of only a fraction of the training parameters.

# 6. Limitations & Conclusion

**Limitations** Our experiments demonstrate that our framework successfully learns families of environments that share either minimal or extensive structure. However, several limitations warrant consideration: (1) MixER's interpretability performance on closely related datasets is inferior to single meta-learners, particularly in scenarios with abundant data availability; (2) the computational demands typically exceed those of individually trained meta-learners, as all experts must remain simultaneously loaded in memory.

**Future Work** Our work establishes foundations for promising research directions beyond the aforementioned limitations. The cluster-expert associations which were observed to *dynamically* shift during training suggest interesting potential for continual learning. Also, exploring the combination of meta-learners with *different* nature or architecture could significantly broaden the usable datasets.

**Conclusion** We integrated traditional ML techniques within deep learning to address the open problem of reconstructing families of dynamical systems with arbitrary relatedness. Through our analysis, we identified the inherent limitations of task-specific meta-learning and proposed as a solution MixER—a Mixture of Experts approach featuring a specialized routing mechanism. Our results demonstrated that while MixER excels when processing highly heterogeneous data with limited amounts of training examples, it conversely underperforms classical meta-learning baselines on datasets exhibiting high degrees of relatedness, with individual experts being exposed to only a fraction of the dataset. Nevertheless, by successfully extending meta-learning from multi-environment DSRs to hierarchies thereof, our findings establish a promising pathway toward domain-agnostic foundational models for scientific applications.

## Impact Statement

This work advances scientific modeling by enabling AI systems to learn from diverse datasets simultaneously. The high computational requirements and complexity of the system could exacerbate research inequity between well-resourced and under-resourced institutions. To address this concern, we open-source our implementation at `https://anonymous.4open.science/r/MixER`, with pre-trained weights optimized for resource-constrained environments to follow.

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

# A. Algorithms & Definitions

## A.1. Llyod's $K$-Means

---
**Algorithm 2** Lloyd's K-Means
---
1: **Require:** $\Xi := \{\xi^e\}^{e\in[E\times F]}$ context vectors
2: $\qquad\qquad \bar{\Xi} := \{\bar{\xi}_m\}_{m\in[M]}$ centroid initialization
3: **if** $\bar{\Xi} = $ Null **then**
4: $\quad \bar{\Xi} \leftarrow $ RandomUniformSample$(M, d_\xi)$
5: **end if**
6: **repeat**
7: $\quad C_m \leftarrow \{\xi \in \Xi : m = \underset{j}{\arg\min} \|\xi - \bar{\xi}_j\|_1\}, \quad \forall m \in [M]$
8: $\quad$ **if** $|C_m| = 0$ **then**
9: $\qquad$ **Return** $\{C_m\}_{m\in[M]},$ Null
10: $\quad$ **else**
11: $\qquad \bar{\xi}_m \leftarrow \dfrac{1}{|C_m|} \displaystyle\sum_{\xi \in C_m} \xi, \quad \forall m \in [M]$
12: $\quad$ **end if**
13: **until** $\bar{\Xi}$ converges
14: **Return** $\{C_m\}_{m\in[M]}, \bar{\Xi}$
---

## A.2. Metric Definitions

We define the relative MSE or relative $L^2$ loss used to perform model selection in several experiments.

$$\text{Rel. MSE} \triangleq \frac{1}{E \times I \times T} \sum_{e=1}^{E} \sum_{i=1}^{I} \sum_{t=1}^{T} \frac{\|x_t^{e,i} - \hat{x}_t^{e,i}\|_2^2}{\|x_t^{e,i}\|_2^2}. \tag{9}$$

To avoid numerical instability in the metric computation, we only consider states $x_t^{e,1}$ with $L^2$ norm grater than $10^{-6}$. Additionally, we consider the TPRMSE (Thresholded Percentage Relative MSE) defined as the proportion of environments in which the Rel. MSE is below a specified threshold $\varepsilon$:

$$\text{TPRMSE} \triangleq \frac{100}{E} \sum_{e=1}^{E} \mathbb{1}_{\{\text{RelMSE}_e < \varepsilon\}}, \tag{10}$$

where:

- $\mathbb{1}_{\{.\}}$ is the indicator function,

- $E$ is the total number of environments available,

- RelMSE$_i$ is the aggregate relative MSE across trajectories in the $e$-th environment.

## B. Datasets

We describe the datasets used in this paper. We begin with the synthetic ODEBench datasets used both to illustrate the limitations of classical meta-learning and those of our MixER in high-data regimes. We follow with the classical synthetic DSR datasets, and finish with real-world EEG data.

### B.1. ODEBench

ODEBench (d'Ascoli et al., 2024) features a selection of ordinary differential equations primarily from (Strogatz, 2018), to which other iconic systems have been added. In total, it boasts 63 definitions of ODE families spanning various regimes: chaotic, periodic, etc., and dimensionality: 1D, 2D, 3D, and 4D. We study 10 of these families, all two-dimensional. First, we describe the data generation process for generating ODEBench-10A (introduced in Table 1), whose training and adaptation trajectories is obtained by adapting the default ODEBench initial conditions and parameters as described below.

The initial conditions for each ODE are generated by interpolating between two reference initial conditions. For each dimension of the ODE, the initial conditions are sampled uniformly between the minimum and maximum values of the two reference conditions. This ensures a diverse set of starting points for the trajectories while maintaining consistency with the ODE's physical or mathematical constraints.

The parameters of the ODEs (e.g., $c_0$, $c_1$) are selected based on predefined reference values. For training and testing, these parameters are varied within a range of 90% to 110% of their reference values. This variation is achieved by creating a grid of parameter values, ensuring a systematic exploration of the parameter space. For adaptation tasks, the parameters are scaled linearly between 80% and 120% of their reference values to simulate environments outside the training domain.

The ODEs are solved using the `solve_ivp` function from the `scipy.integrate` module, which the Runge-Kutta method of order 4(5) (RK45). This method is a widely used numerical integrator for solving initial value problems due to its balance between accuracy and computational efficiency. The evaluation time step for reporting $\Delta t$ is determined by dividing the time horizon $T$ by the number of steps (fixed to 100 across all families), ensuring a consistent resolution across all simulations. We focus on ODEs that display a *periodic* behavior, and the time horizon is chosen so as to observe at least one full oscillation.

The dataset is divided into four distinct splits: train, test, adaptation train, and adaptation test. The number of environments and initial conditions for each split is summarized in the table below.

*Table 5.* Data splits and their characteristics for ODEBench-10A. Similar attributes apply to ODEBench-10B as per Table 1.

| Split | Environments | Initial Conditions | Description |
|---|---|---|---|
| Train | 5 | 4 | Used for training models. |
| Test | 5 | 32 | Used for evaluating model performance. |
| Adaptation Train | 1 | 1 | Used for fine-tuning context vectors. |
| Adaptation Test | 1 | 32 | Used for evaluating fine-tuned contexts. |

The following table provides a detailed description of the ODE families used in the dataset. Each ODE is identified by an ID, and its analytical definition, time horizon, initial conditions, and parameters are listed.

*Table 6.* ODE identifiers and definitions from (d'Ascoli et al., 2024), along with custom parameters, initial conditions, and time horizon values. The custom values are used to generate ODEBench-10A and ODEBench-10B.

| ID | Family Name | Equation | Parameters | Initial Values | Time Horizon |
|---|---|---|---|---|---|
| 24 | Harmonic oscillator without damping | $\begin{cases} \dot{x}_0 = x_1 \\ \dot{x}_1 = -c_0 x_0 \end{cases}$ | $c_0 = 2.1$ | $[0.4, -0.03]$ $[0.0, 0.2]$ | 10 |
| 25 | Harmonic oscillator with damping | $\begin{cases} \dot{x}_0 = x_1 \\ \dot{x}_1 = -c_0 x_0 - c_1 x_1 \end{cases}$ | $c_0 = 4.5$ $c_1 = 0.43$ | $[0.12, 0.043]$ $[0.0, -0.3]$ | 8 |

| | | | | | |
|---|---|---|---|---|---|
| 28 | Pendulum without friction | $\begin{cases} \dot{x}_0 = x_1 \\ \dot{x}_1 = -c_0 \sin(x_0) \end{cases}$ | $c_0 = 0.9$ | $[-1.9, 0.0]$ $[0.3, 0.8]$ | 15 |
| 32 | Damped double well oscillator | $\begin{cases} \dot{x}_0 = x_1 \\ \dot{x}_1 = -c_0 x_1 - x_0^3 + x_0 \end{cases}$ | $c_0 = 0.18$ | $[-1.8, -1.8]$ $[-2.8, 1.0]$ | 5 |
| 34 | Frictionless bead on a rotating hoop | $\begin{cases} \dot{x}_0 = x_1 \\ \dot{x}_1 = (-c_0 + \cos(x_0)) \sin(x_0) \end{cases}$ | $c_0 = 0.93$ | $[2.1, 0.0]$ $[-1.2, -0.2]$ | 20 |
| 35 | Rotational dynamics of an object in a shear flow | $\begin{cases} \dot{x}_0 = \frac{\cos(x_0)}{\tan(x_1)} \\ \dot{x}_1 = \sin(x_0)(c_0 \sin^2(x_1) + \cos^2(x_1)) \end{cases}$ | $c_0 = 4.2$ | $[1.13, -0.3]$ $[0.7, -1.7]$ | 5 |
| 37 | Van der Pol oscillator (standard form) | $\begin{cases} \dot{x}_0 = x_1 \\ \dot{x}_1 = -c_0 x_1 (x_0^2 - 1) - x_0 \end{cases}$ | $c_0 = 0.43$ | $[2.2, 0.0]$ $[0.1, 3.2]$ | 15 |
| 38 | Van der Pol oscillator (simplified form) | $\begin{cases} \dot{x}_0 = c_0 \left( -\frac{x_0^3}{3} + x_0 + x_1 \right) \\ \dot{x}_1 = -\frac{x_0}{c_0} \end{cases}$ | $c_0 = 3.37$ | $[0.7, 0.0]$ $[-1.1, -0.7]$ | 15 |
| 39 | Glycolytic oscillator | $\begin{cases} \dot{x}_0 = c_0 x_1 + x_0^2 x_1 - x_0 \\ \dot{x}_1 = -c_0 x_0 + c_1 - x_0^2 x_1 \end{cases}$ | $c_0 = 2.4$ $c_1 = 0.07$ | $[0.4, 0.31]$ $[0.2, -0.7]$ | 4 |
| 40 | Duffing equation | $\begin{cases} \dot{x}_0 = x_1 \\ \dot{x}_1 = c_0 x_1 (1 - x_0^2) - x_0 \end{cases}$ | $c_0 = 0.886$ | $[0.63, -0.03]$ $[0.2, 0.2]$ | 10 |

The ODEBench-2 dataset is a subset of the original ODEBench-10 datasets, focusing on two specific systems: the Harmonic Oscillator with Damping (ID 25) and the Rotational Dynamics of an Object in a Shear Flow (ID 35) from Table 6. A few changes where made to emphasize the differences in dynamics behaviour between the two families. Those changes are summarized in Table 7.

*Table 7.* Parameter, initial condition, and time horizon values for ODEBench-2.

| ID | Parameters | Initial Values | Time Horizon |
|---|---|---|---|
| 25 | $c_0 = 0.4$ | $[0.1, 0.1]$ $[0.0, -0.3]$ | 5 |
| 35 | $c_0 = 6.0$ | $[1.13, -0.3]$ $[0.7, -1.7]$ | 5 |

### B.2. LV, GO, and SM

The Lotka-Volterra (LV), Glycolytic Oscillator (GO), and Sel'kov Model (SM) have been the subject of extensive studies these past years. A complete description of each dataset along with the generation processes is provided in (Yin et al., 2021; Kirchmeyer et al., 2022; Nzoyem et al., 2025). For our use case, we download the data from the Gen-Dynamics repository (Nzoyem et al., 2025).

### B.3. Synthetic Control

The Synthetic Control Chart Time Series (SCCTS) dataset is a collection of synthetically generated control charts, designed for time series clustering and classification tasks. The dataset contains 600 time series instances, each comprising 60 time steps, and is divided into six distinct classes: Normal, Cyclic, Increasing Trend, Decreasing Trend, Upward Shift, and Downward Shift. The dataset has been used in prior research to explore time series similarity queries and control chart pattern recognition. Key references include works by (Alcock, 1999) on feature-based time series similarity, and (Pham & Chan, 1998) on neural network-based control chart recognition.

The primary task associated with this dataset is clustering, with a focus on evaluating the performance of time series clustering algorithms. The dataset is particularly useful for testing algorithms that go beyond the Euclidean distance, as certain class pairs are often misclassified using traditional distance measures. For instance, Derivative Dynamic Time Warping (DDTW) (Keogh & Pazzani, 2001) has been shown to achieve better clustering results compared to Euclidean distance. The raw dataset was downloaded from `https://www.timeseriesclassification.com/description.php?Dataset=SyntheticControl`.

### B.4. Epilepsy2

The Epilepsy2 dataset comprises single-channel electroencephalogram (EEG) measurements collected from 500 subjects (Andrzejak et al., 2001; Zhang et al., 2022). For each subject, brain activity is recorded over a duration of 23.6 seconds, then partitioned and shuffled, resulting in 11,500 examples (80 for training, and 11,420 for testing), each spanning 1 second and sampled at 178 Hz.

The raw dataset downloaded from `https://www.timeseriesclassification.com/description.php?Dataset=Epilepsy2` includes five classification labels corresponding to different subject states or measurement locations: eyes open, eyes closed, EEG from a healthy brain region, EEG from a tumor-affected region, and seizure episodes. For binary classification as performed in Section 4.3, the first four classes were merged into a single "no seizure" class, while the seizure episodes were retained as the "seizure" class. The training set is balanced, containing 40 seizure and 40 non-seizure samples, whereas the test set is imbalanced, with 19.79% seizure and 80.21% non-seizure samples.

# C. Implementation Details

We describe the implementation of our MixER framework through the lens of its routing and its hyperparameters. We also present the baselines and the changes we made to fit them within our framework.

## C.1. Context-Based Routing

In our framework, the only way the gating network influences the output is via the logits it produces for routing (see Figure 3). We effectively eliminate the final aggregations so that the expert can be used on its own outside the MoE layer. This has adverse consequences however, in that the gating doesn't impact the output enough to receive high gradients. While this is generally solved with our clustering mechanism, we find that two mechanisms improve the clustering when the relatedness of families is minimal:

1. **Context Splitting.** The router splits the contexts $\xi$ into $m$ equal-length pieces $\{\xi_m\}_{m\in[M]}$ before feeding them to the experts. This means each experts only ever sees a specific portion of the contexts. We apply this only on the IVPs tested in this paper.

2. **Context Shifting.** Each expert is augmented with a single floating point offset, by which the inputted contexts are shifted before usage. Again, with shifts the overall mean of the contexts received by the experts, further facilitating clustering. We apply this to all experiments conducted in this paper.

## C.2. Core Baseline Methods

With the exception of CAVIA, we perform a custom implementation of several baselines and incorporate them within our MixER layer.

• **CAVIA (Zintgraf et al., 2019)**   is a concatenation-based meta-learning approach that improves on the seminal (Finn et al., 2017) by optimizing parameter-specific context vectors in its inner loop. Within the model $G_\theta$, pre-processing of $\xi^e$, $z_{t-1}$, and $x_{t-1}^e$ may be performed before concatenation and processing within a main network.

• **Neural Context Flows (Nzoyem et al., 2025)**   use a first-order optimization procedure coupled with contextual self-modulation to share information between environments, thus encouraging the formation of clusters and improving generalization. We use 2nd order Taylor expansion resulting in NCF-$t_2$. Its model $G_\theta$ processes inputs like in CAVIA.

• **CoDA (Kirchmeyer et al., 2022)**   is aimed at initial value problems and leverages a linear hypernetwork to generate environment-scpefic weights of the root (main) network based on context vectors.

• **GEPS (Koupaï et al., 2024)**   improves on CoDA's scalability by performing low-rank adaptation on MLP and CNN weights, conditioned on context vectors. In our implementation, we use Xavier initialization (Glorot & Bengio, 2010) for the $A$ and $B$ matrices, and we initialize the contexts at 0.

• **hier-shPLRNN (Brenner et al., 2024)**   is a fast sequence-to-sequence shallow Recurrent Neural Network meta-learner. Similar to CoDA, subject-specific weights are generated with a linear hypernetwork. We set the width of its single hidden layer to 16. Our setting does not require any encoders to map $x$ to $z$, which live in the same space. We set the initial $z_0 = 0$.

## C.3. Main Hyperparameters

**Training**   All the experts in the MixER are initialized with the same seed. Across our experiments, the batch size is the expected number of environments per expert, i.e. $E/M$. We use the AdaBelief optimizer (Zhuang et al., 2020) for both contexts and weights. Adaptation to new environments is performed on a sequential one by one basis, except on the Epilpesy2 dataset which considers batches of size 571.

**Gating Update**   In our proximal alternating minimization, we performed up to 500 outer iterations, and 12 inner iterations of both weights $\Theta$ and contexts $\Xi$, with the gate updated every time either are updated. We upper bound the number of iterations in $K$-means to 20 (see Algorithm 2), and we set the convergence tolerance to $10^{-3}$ and the noise standard deviation to $10^{-4}$ (see Algorithm 1).

**Architectures**    On problems using ODEBench, we use a 3-layer MLP of width 64 as the main network. For NCF, we use shallow context and data networks or depth 1, each with outputs of size 32. We use the Swish activation (Ramachandran et al., 2017) throughout, except with the hier-shPLRNN where we use ReLU activations. On other IVP problems, we adjust the width of the main layer so that the active parameter count (equal to the number of parameters in *one* expert (Jiang et al., 2024)) matches the baselines.

**Software**    We use JAX (Bradbury et al., 2018) and its differentiable programming ecosystem (Nzoyem et al., 2023). Specifically, we use diffrax and its Tsit5 solver to integrate differential equations (Kidger, 2022), with all neural networks implemented with Equinox (Kidger & Garcia, 2021).

**Hardware**    Depending on the experiment, our model was trained on a workstation fitted with a NVIDIA 4080 GPU with 16GB VRAM memory, and a supercomputer containing four NVIDIA GH200 GPUs with 480GB total memory. We aimed for quick training times, with hier-shPLRNN being by far the faster to train in less than 5 minutes on both Epilepsy and SCCTS datasets. It took CoDA around 20 minutes, GEPS 30 minutes, and finally NCF 25 minutes to complete 500 outer steps on the largest ODEBench-10B dataset.

# D. Qualitative Results

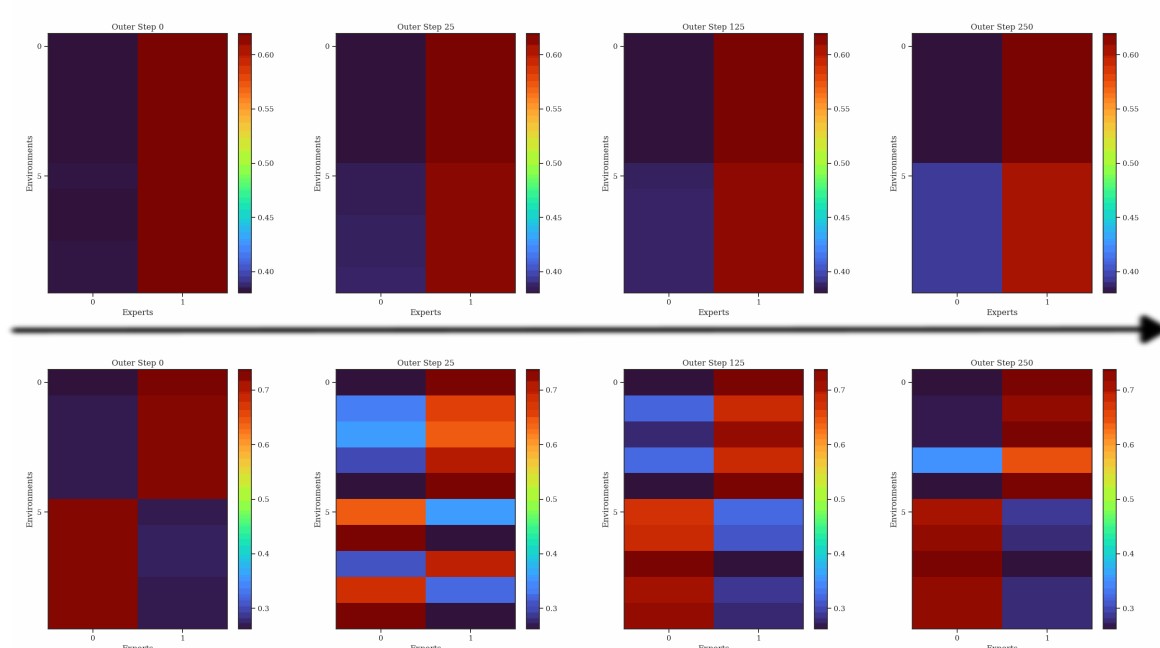

*Figure 8.* Visualisation of the clustering heatmap as the training progresses on ODEBench-2. The four columns correspond to outer training steps 0, 25, 125, and 250 respectively (from left to right). (Top) Naive mixture of two GEPS models with gating updates via vanilla gradient descent. (Bottom) MixER and least-squares-based gating update.

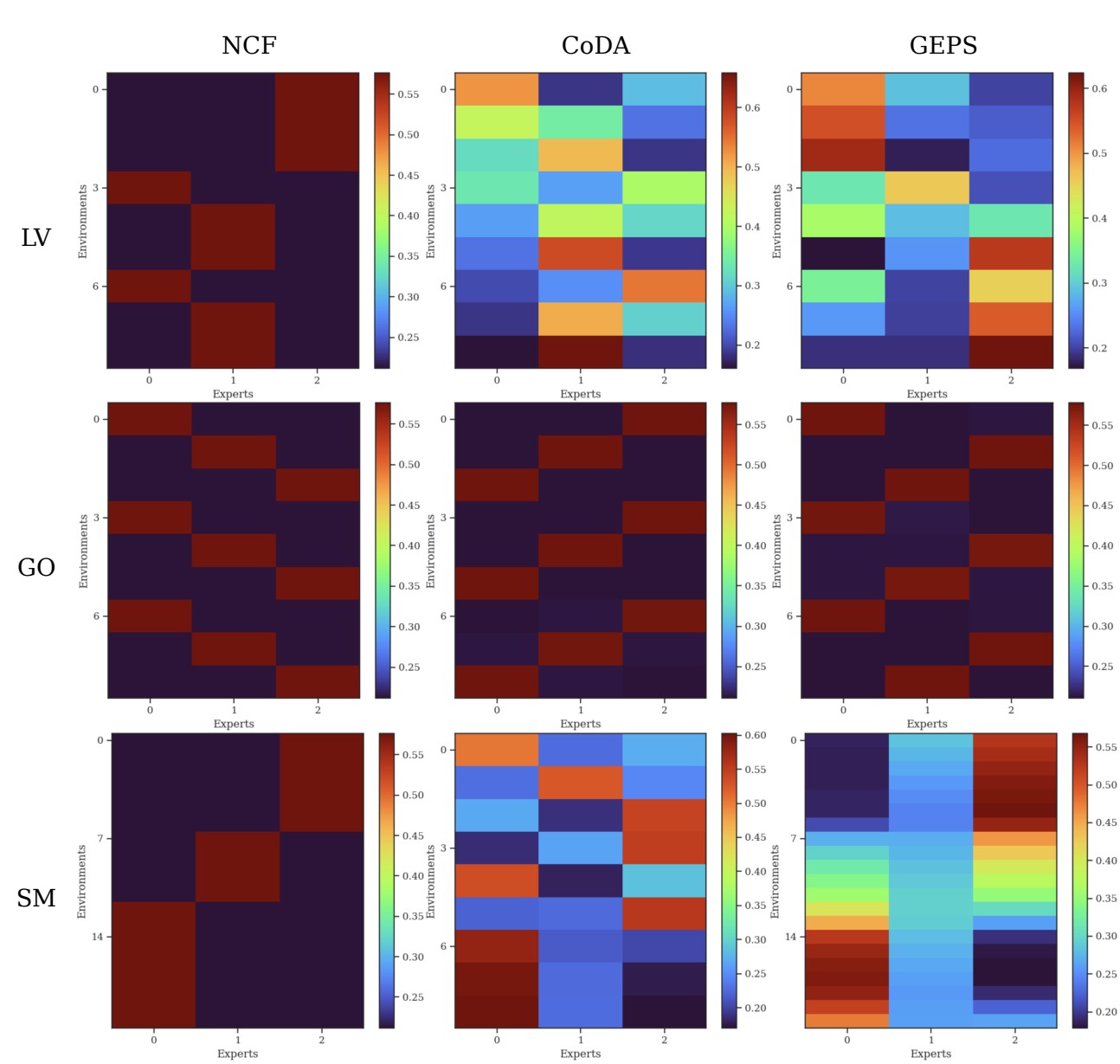

*Figure 9.* Heatmaps of the gating values of MixER with 3 experts on three classical meta-learning datasets: LV, GO, and SM.

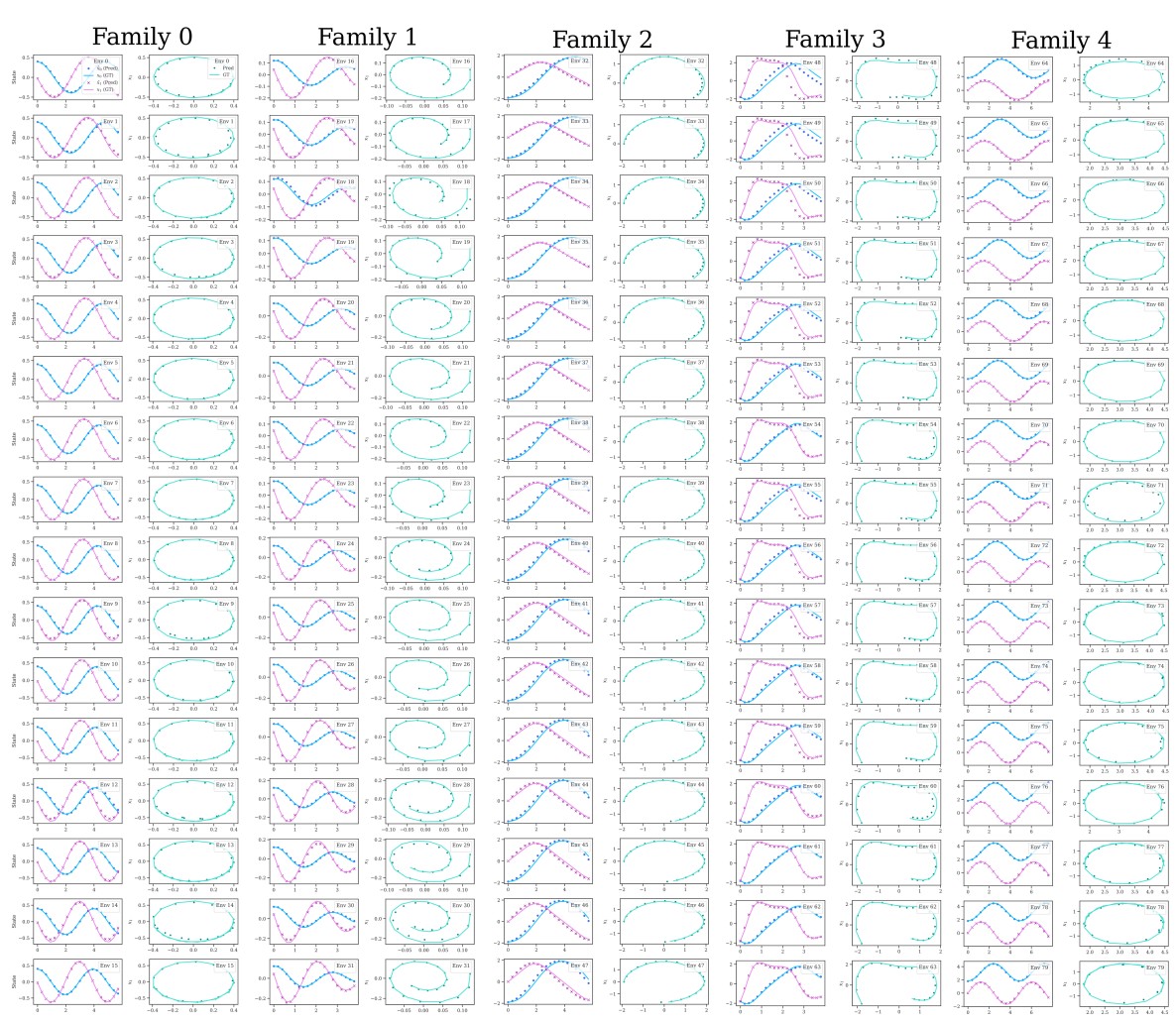

*Figure 10.* Visualization of a single testing trajectory and the phase space within the *first* 5 families with 10 expert GEPS meta-learners on the large ODEBench-10B dataset.

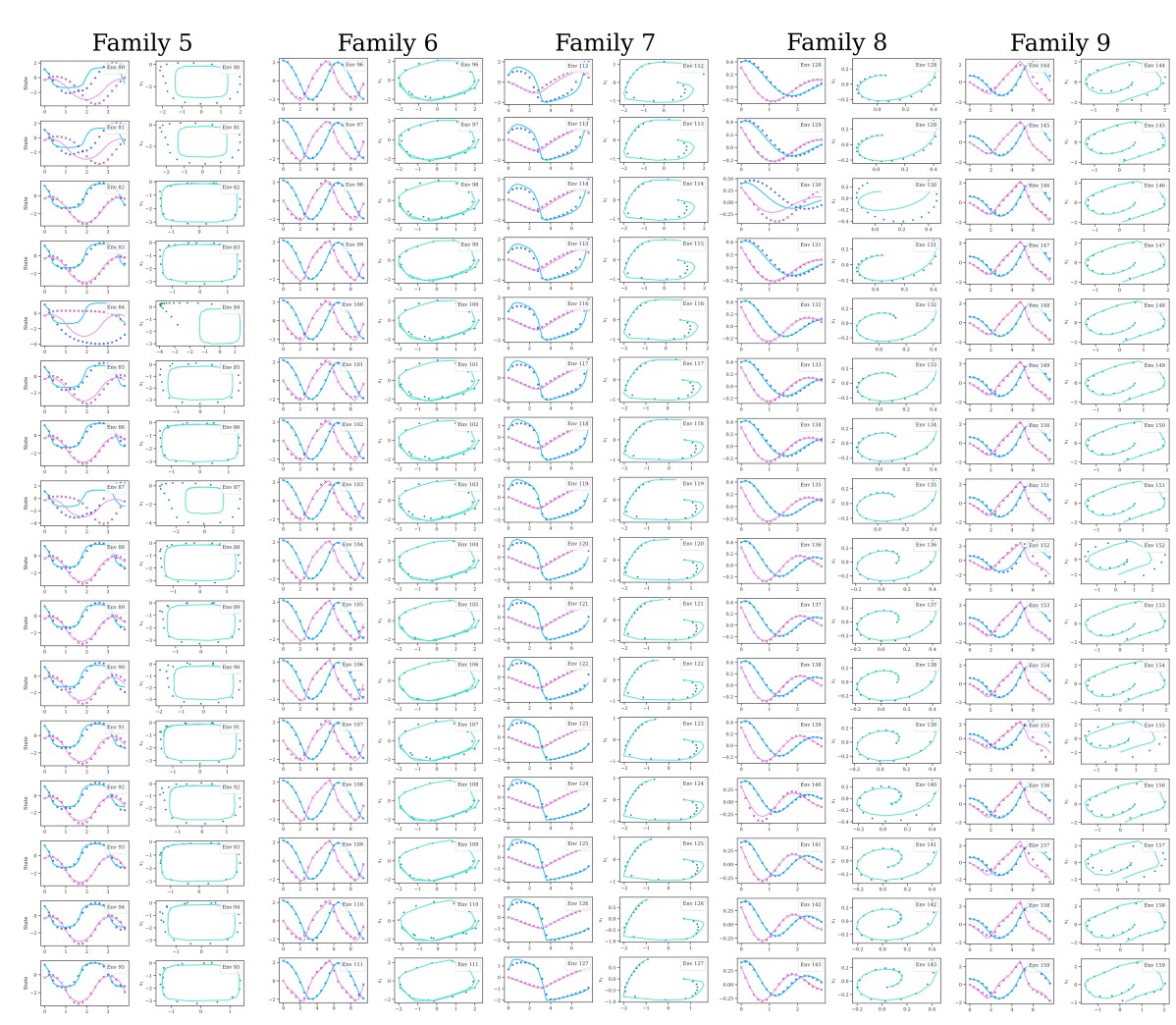

*Figure 11.* Visualization of a single testing trajectory and the phase space within the *last* 5 families with 10 expert GEPS meta-learners on the large ODEBench-10B dataset.

