# OpenReview forum: "Towards Foundational Models for Dynamical System Reconstruction: Hierarchical Meta-Learning via Mixture of Experts"
_ICML.cc/2025/Conference — Submitted to ICML 2025_

### Official Review · Reviewer_6AnX · 2025-03-13

**Overall Recommendation:** 2

**Summary:**

The authors introduce a novel Mixture of Experts (MoE) architecture for Dynamical Systems Reconstruction (DSR): Mixture of Expert Reconstructors (MixER). MixER employs a top-1 MoE strategy with a custom routing mechanism that enables unsupervised clustering and meta-learning across different DS. The manuscript explores benefits and limitations of the approach by applying the method to various DSR meta-learning backbones across a platitude of different datasets.

**Claims And Evidence:**

“We propose an effective unsupervised routing mechanism for MoEs to collectively learn dynamical systems with various degrees of relatedness.” (p. 2): In my opinion, this claim  lacks convincing evidence. The proposed method rarely outperforms baseline setups, i.e. MixER-$1$ and MixER-$N^{\dagger}$ with naive routing mechanism. Also, MixER seems to *prefer* highly heterogenous data, i.e. where families of environments are clearly different and struggles in cases of high relatedness (as pointed out by the authors).

**Essential References Not Discussed:**

I am not aware of any essential references that need mentioning.

**Experimental Designs Or Analyses:**

All experiments and analyses seem valid.

**Methods And Evaluation Criteria:**

Methods and evaluation criteria seem mostly appropriate. However, including a benchmark dataset that encompasses chaotic dynamical systems (e.g. [1]) would further substantiate the method. Of course I acknowledge the real-world data (Epilepsy-2), which is a challenging dataset with signatures of chaotic dynamics. However, I think it would make sense to first validate the method on clear benchmarks, which do not necessarily face the problem of high ambiguity in the data.

**Other Comments Or Suggestions:**

**Other Comments**:
- Figure 5 axis labels and ticks are quite small and should be increased in size. I also don’t think Figure 5 bottom is strictly necessary, it does not really provide more information than the top row in my opinion. It would be more interesting to see heatmaps of MixER-20 (see also questions)

- Figure 6 legend and axis labels are extremely small and hard to parse.

**References**

[1] Gilpin, W. (2023). Model scale versus domain knowledge in statistical forecasting of chaotic systems. Physical Review Research, 5(4), 043252.

 [2] Hess, F. et al. (2023). Generalized Teacher Forcing for Learning Chaotic Dynamics. In International Conference on Machine Learning (pp. 13017-13049). PMLR

**Other Strengths And Weaknesses:**

**Strengths**: The paper tackles an unaddressed problem: Extracting DSR models from heterogeneous data using a potentially highly interpretable approach and it’s great that the method is tested on multiple benchmarks, real-world data and across different meta-learning backbones.

**Weaknesses**: It is not clear under which circumstances (e.g. dataset + backbone combination) MixER with M > 1 experts consistently improves overall performance and in which cases performance is degraded, irrespective of the actual underlying number of families in the data at hand.

**Questions For Authors:**

1. How are experts selected if $M > F$ ? I.e. how interpretable is the architecture when the number of experts exceeds the number of families? Does MixER naturally converge to only use $\approx F$ experts or does it in a sense “overfit” and potentially split environments from one family into two separate families? In that context, it would be nice to see the heatmap in Fig. 5 for MixER-20, too, at least for GEPS.
2. Do the authors have more input on why MixER seems to prefer GEPS over NCF and CoDA as a backbone? Why does MixER lead to more specialization in case of GEPS (Tab. 3, Fig. 5 top)?
3. Can the authors comment on the scalability of MixER in terms of number of environments and experts? Computing losses of all experts on all environments seems extremely demanding, especially if the dynamics under consideration become high-dimensional and the dynamics backbones become more computationally demanding themselves. I think this point is especially important in the context of foundation models.
4. The ODEBench-X mainly encompasses dynamics of fixed point or cyclic dynamics. How does the algorithm (with an appropriate backbone) fare on a dataset of families of chaotic dynamics, e.g. the dataet used in [1]? Mere MSE losses might not be a suitable measure for expert routing, based on the fact that MSE is not necessarily a sufficient measure of DSR in those settings [1,2].
5. “Clustering and routing analysis (Figure 9) shows that MixER logically partitions datasets into three subsets, but this partitioning limits each expert meta-learner’s exposure to the full dataset, potentially explaining the performance degradation despite clear cross-environment commonalities.” → But does this not show a fundamental problem with the approach? If this is the case, then what is the benefit of MixER in this setting? If we know the number of families, we’d seem to be better off by training meta-learners on those families separately? I think what MixER misses is some evidence that if the number of experts is larger than the number of families in the dataset, MixER converges to use only the minimum necessary number of experts to explain the data. This way we would learn something fundamental about the data (if we do not know the families a priori, which is the general setting considered in the paper).
6. Validataion loss dynamics of MixER seem quite irregular and "spiky" (Fig. 2 bottom). Can the authors comment on this?

*I am happy to increase my score to accept if the authors can address my concerns*.

**Relation To Broader Scientific Literature:**

The paper uses the concept of Mixture of Experts (MoE) to introduce a novel routing method, MixER, to learn across diverse families of dynamical systems which is a mandatory concept when thinking of foundational models in this field. Current research mainly deals with meta-learning across different environments, which generally span different parameter settings (possibly across bifurcations) of one dynamical system but are generally unable to learn across (potentially) unrelated systems.

**Theoretical Claims:**

There are no theoretical claims and therefore no proofs.

---

> ### Author Rebuttal · Authors · 2025-03-30
>
> We sincerely appreciate the reviewer’s thorough and insightful evaluation. Your recognition of MixER as a novel approach is encouraging. We are grateful for the acknowledgment that MixER addresses an important and previously unaddressed problem in DSR and that our experiments are well-executed across multiple benchmarks and real-world datasets. We also appreciate your comments on the relevance of our work to foundational models and your endorsement of our experimental design and evaluation criteria.
>
> ---
> ### Q1. Expert Selection and Interpretability
>
> When $M>F$, MixER does not impose strict constraints on the number of experts used. While this may lead to splitting environments from the same "family", it may reflect nuanced differences rather than overfitting.
>
> As suspected by the reviewer, on ODEBench-10B, MixER splits the 10 families into subfamilies so that none of the 20 experts are unused (see https://anonymous.4open.science/r/MixER/assets/odebench_heatmaps_20M.png). A similar pattern is visible in Figure 9.
>
> Our new experiment on chaotic systems (see Q4) reveals that up to three experts might be necessary, even when only $F=2$ families were envisioned. With $M=4$, the model converges to using only three experts. Through Algorithm 2, MixER allows for empty K-means clusters, effectively sidelining unused experts (https://anonymous.4open.science/r/MixER/assets/lorentz_heatmaps.png).
>
> In summary, while MixER may split our tentative families, it captures the underlying structure in the data. While we cannot guarantee when the system converges to fewer experts, we suspect this behavior depends on the dataset and the backbone. Future Work on MixER will investigate theoretical guarantees, and devise synthetic datasets that expose this behavior.
>
> ---
> ### Q2. Preference for GEPS as a Backbone
>
> We think GEPS is preferred over NCF and CoDA due to its greater flexibility and expressiveness in capturing shared knowledge. Further analysis of each backbone's properties could provide deeper insight, and we will include this in the Future Work section.
>
> ---
> ### Q3. Scalability Considerations
>
> Efficient approximations and batching strategies (batch size $B\ll E$) mitigate scalability challenges. Plus, the gate is updated only via **forward** passes, and no backpropagation is performed. This leads to very fast updates. In practice, MixER incurs roughly a 20% slowdown over naive MoE (OdeBench-2):
>
> ||CoDA|GEPS|
> |---|---|---|
> |MixER|933 s|449 s|
> |Naive MoE|790 s|375 s|
>
> We acknowledge that computational costs can increase when dynamics are demanding. We will highlight this as a limitation in Section 6.
>
> ---
> ### Q4. Performance on Chaotic Dynamics
>
> We appreciate and will include the references [1] and [2].
>
> For evaluation on chaotic dynamics, we merged noisy Lorentz63 and noisy Lorentz96 datasets from [3]. Lorentz63 trajectories were padded with 0 to match Lorentz96’s dimensionality (10). The optimization loss was Gaussian negative likelihood, used for expert routing. In addition to the $L_2$ evaluation loss, we considered the Hellinger distance $D_H$ for long-term dynamics from [3].
>
> Although we expected two experts hier-shPLRNN [3] to be ideal, we tested MixER with up to four experts. Looking at $D_H$, three experts proved optimal, with the fourth unused in MixER-4 (see link in Q1). Two experts handled most environments, while the third handled outlier cyclic trajectories.
>
> ||MixER-1|MixER-2|MixER-3|MixER-4|
> |---|---|---|---|---|
> |$L_2$|4.557e-03|1.793e-03|1.442e-03|1.038e-03|
> |$D_H$|5.610e-01|5.555e-01|5.543e-01|5.544e-01|
>
> [3] Brenner et al., ICLR 2025, Learning Interpretable Hierarchical Dynamical Systems Models from Time Series Data
>
> ---
> ### Q5. Fundamental Limitations and Benefits of MixER
>
> We agree that MixER does not offer much benefit on classical DSR datasets, especially if SoTA performance is the primary goal. (In fact, we included these datasets specifically to highlight this limitation of MoE approaches.)
>
> That said, as noted in Q1 and Q4, MixER can uncover fundamental information in the data. We recommend using a small number of experts to minimize computational costs.
>
> ---
> ### Q6. Validation Loss Dynamics
>
> In Figure 2 (bottom), this is due to abrupt shifts in gating function assignments: environments being wrongly assigned, then corrected in the next step. This is partly due to adding noise (Algorithm 1, line 23) before performing least squares regression, done to prevent the system from getting stuck in suboptimal configurations.
>
> ---
> ### Q7. Weakness: Under which Circumstances does MixER Improve Performance?
>
> Figure 2, Table 2, and Table 3 show diminishing returns on ODEBench2, ODEBench-10A, and ODEBench-10B, respectively. MixER performs best in low-data regimes and when families are highly distinct, which we believe will be the dominant setting for foundational DSR models.
>
> Additionally, we appreciate the reviewer’s comments on Figures 5 and 6 and will update them accordingly if accepted for publication.

---

> > ### Comment · Reviewer_6AnX · 2025-04-07
> >
> > I thank the authors for the additional experiments and clarification during the rebuttal phase.
> >
> > **Q1 + Q4**
> > Thanks. It makes total sense that MixER separates different dynamics, rather than separating the environments based on the ground truth underlying system, i.e. it makes much more sense for the algorithm to route cyclic dynamics to a separate expert, even though this does not reflect that the origin of the data is the same set of ODEs. That MixER-4 only uses 3 experts is nice to see.
> > It is also great to see that the authors considered long-term dynamics measures such as $D_H$. However, the results puzzle me - did the authors correctly apply the measure? In the original publication [1] the results on the Lorenz datasets in terms of $D_H$ is an entire order of magnitude smaller. It also features no variance across different MixER settings, which suggests that all combinations produce similarly bad (good) results? What is the standard deviation across trajectories?
> >
> > **Q2**
> > I see, thanks. To me it seems quite important that the backbone maximally captures shared knowledge, as this would most benefit MixER by minimizing the required experts. That said, I do think backbone performance is a separate topic and does not influence the significance of the manuscript.
> >
> > **Q3**
> > I see, however, the approach still needs $M_B \times E_B$ forward pass evaluations for each batch, correct? Paired with the premise that the approach aims at foundational models, this scaling might be prohibitive since, as of today, foundational models are trained on vast amounts of data, and hence generally need larger batch sizes for optimal convergence. I think the manuscript would hence benefit from a more rigourous runtime complexity analysis (if at least in the appendix).
> >
> > **Conclusion**
> > After also reading the other reviews and rebuttal answers, my conclusion is that the paper isn’t quite ready yet. Specifically, if the authors aim to keep the “foundational models” aspect, I suggest a more rigorous runtime and scaling analysis.  Also, the method only seems to increase performance over baselines in very specific settings (highly heterogeneous data in terms of families, little data in terms of trajectories/environments). That MixER converges to a reasonable number of experts in the rebuttal experiment with the Lorenz datasets is a nice result, but it does not convince me to lean towards accept. I will hence keep my score.

---

> > > ### Author Response · Authors · 2025-04-07
> > >
> > > Dear Reviewer,
> > >
> > > Thank you for acknowledging our rebuttal efforts and taking the time to provide additional comments.
> > >
> > > **Q1+Q4**. We have leveraged the code from [3] to compute the Hellinger distance $D_H$, and we have no doubt their implementation is accurate. Given your statement that you would maintain your score regardless—despite previously implying openness to change contingent on a thorough rebuttal of the concerns you **previously** had—we see limited value in performing additional experiments to include standard deviations at this time. That said, thank you for pointing that out.
> > >
> > > **Q3**. Thank you for this helpful suggestion. We agree that examining the influence of the batch size $B$ on both performance and runtime would strengthen the paper. We will include this analysis in the appendix.

---

### Official Review · Reviewer_p7Cj · 2025-03-13

**Overall Recommendation:** 1

**Summary:**

Authors consider the following system reconstruction formulation: given D-dimensional time series, or measurements, of length T, the goal is to reconstruct the process generating this data. The series are grouped into E distinct environments, which are grouped into F families. It is assumed that the measurements within the same environment are closely related, environments within the same family may have higher-order relationships, and the relationship between families can be arbitrary.

Authors highlight two families of DSR models for solving this problem: flat (seq2seq and IVP) and hierarchical, or contextual (hypernet- and concatenation-based). The paper focuses on the second family and studies the question of optimally clustering environments (into families) in order to maximise performance of the underlying contextual models.

To do this, the paper proposes MixER: Mixture of Expert Reconstructors – a sparse top-1 Mixture-of-Experts (MoE) which comprises M contextual DSR model. MixER’s main innovation is in the fact that it uses E environment-conditioned context vectors for gating, and its custom gating update algorithm, which uses K-means clustering and least squares optimization, instead of gradient descent.
The router is a linear mapping from context vectors into M logits (argmax of which is the selected expert), and its parameters update operates in 4 stages:

1. K-means clustering for context vectors, K := M;
2. Calculating median loss for each of M^2 expert-cluster pairs;
3. Greedy pairing of experts and clusters according to the median losses;
4. Least-squares optimisation with one-hot targets derived at step 3.

Authors conduct a set of experiments to showcase strengths and limitations for MixER. They extract three subsets of the ODEBench dataset, each of which has a varying number of environments and families.

Firstly, they demonstrate limitations of existing DSR models, namely NCF, CoDA, and GEPS, on the smallest of three ODEBench-2 task (F=2, E=5) of reconstructing two 2-dimensional ODEs. Additionally, they show that naive MoEs also fail to solve the problem, as opposed to the MixER approach, which is able to generalise to the validation dataset.

Secondly, they report MixER-M models with M experts performance during training and adaptation, claiming that one of two modifications of MixER-10 outperforms MixER-1 in ODEBench-10A (F=10, E=5), but underperforms compared to the baselines on ODEBench-10B (F=10, E=16).

Finally, the paper investigates feature interpretability and downstream clustering on two time-series datasets, Epilepsy2 and SCCTS, showing that the clustering labels derived by MixER are highly correlated with the ground truth labels for SCCTS, however not so much for Epilepsy2. Additionally, in spite of having extra model capacity, MixER underperforms on both of these datasets compared to the baselines.  Authors hypothesize that the reason for that may be dataset’s inherent noise and close relationship between presented environments.

**Claims And Evidence:**

Authors make several claims in the paper, however, in my opinion, they fail to provide clear evidence for most of them. I will comment on the main three claims made in the paper.

1. L78-82, “We claim that strategic combination of contextual meta-learners enables simultaneous reconstruction across all families while preserving rapid adaptation capabilities, obviating the need for manual dataset partitioning prior to meta-learning on each subset”  – while this may generally be true, the paper doesn’t provide strong evidence in support of this. Moreover, the proposed method uses several manually designed heuristics in its gating mechanism (k-means, greedy pairing, argmax for top-1), whose efficiency is unclear compared with any manual/fully heuristic dataset partitioning. Also, the results from section 4.1 show that MixER struggles with adaptation, hence the claim remains unproven.

2. L100-104, “We identify a fundamental limitation of gradient descent when routing contextual information to DSR models, which slows down expert specialization when training MoEs” – this claim is purely based on the experimental results in section 2.2, which, firstly, used a small dataset and, secondly, if my understanding is correct, invalid experimental protocol (described below). I also suspect that the baselines were not properly tuned, e.g. authors claim that the naive MoE sends all inputs to the same expert, which can be addressed by tuning load balancing loss.

3. L184-185, “... our framework eliminates the need for importance or load balancing terms in the loss function.” – this claim remains unaddressed in the text. From the algorithm description it follows that the derived clusters can be of arbitrary size (as also demonstrated in Figure 4), and, although Figure 8 suggests that the clusters can be equally sized, there are no guarantees for this, and I’d expect this to not hold in practice in general.

**Essential References Not Discussed:**

I am not aware of any.

**Experimental Designs Or Analyses:**

My main concern is that the authors don’t mention anywhere that all compared models use the same number of parameters, which, apparently, means that they don’t? If it’s the case, then all the experiments in the paper are not invalid due to the incorrect comparisons with the a-priori weaker baseline.

Section 2.2. with the motivational example doesn’t state how many experts are employed in the MoE models. Since the dataset is very small, I wonder how authors chose this number and how they ensure no overfitting? I suspect the latter happens for the MixER model and doesn’t happen for the naive MoE because of its suboptimal hyperparameters.

Another evidence for MixER overfitting is poor results for the adaptation mode in section 4.1.

Why do authors report train-set MSEs in Tables 2 and 3? Is it a typo? If not, what are the numbers on the corresponding test sets?

**Methods And Evaluation Criteria:**

The benchmarks are suitable to investigate the problem at hand.

**Other Comments Or Suggestions:**

Figure 4 shows two clusters of different sizes (apparently, 3 and 5), however the derived Y labels have a 4:4 split. Is it a typo, or is there some additional procedure that ensures equal split between labels? I haven’t found it in the paper.

Some references are incomplete or incorrect, e.g sequence-to-sequence references do not include the [1], and the MLP references don’t mention [2].

[1] Sutskever, Ilya, Oriol Vinyals, and Quoc V. Le. "Sequence to sequence learning with neural networks." Advances in neural information processing systems 27 (2014).

[2] Rosenblatt, Frank (1958). "The Perceptron: A Probabilistic Model For Information Storage And Organization in the Brain". Psychological Review. 65 (6): 386–408.

**Other Strengths And Weaknesses:**

Overall, I am not convinced that experimental results are strong enough to claim that the proposed method is stronger than the baselines. Most of the reported results (except for the motivational example) contain quite mixed numbers, and MixER is subpar to the baseline in most comparisons. Authors include a good analysis of the limitations, however no significant strengths are successfully demonstrated.

**Questions For Authors:**

Please consider addressing my comments from above, which are essentially questions about the paper.

**Relation To Broader Scientific Literature:**

The paper combines the ideas of hierarchical meta-learning and the mixture-of-experts technique. It is an attempt to combine the MoE with K-means clustering in order to train domain-specific experts, where domains form the highest level (family) in a 3-level hierarchy (measurement, environment, family).

**Theoretical Claims:**

No theoretical claims are made in the paper.

---

> ### Author Rebuttal · Authors · 2025-03-31
>
> We thank the reviewer for reviewing our work and for providing thorough comments. We find your summary of our work especially pleasing to read. Thank you for noting the combination of the ideas of hierarchical meta-learning and the mixture-of-experts technique that our method leverages. We are happy our attempt to combine the MoE with K-means clustering in order to train domain-specific experts was worth mentioning. Furthermore, thank you for noting the relevance of our supplementary material, which contains valid descriptions of the used datasets and classic algorithms, visualizations for the experimental results, as well as some MixER implementation details.
>
> ---
> ### Q1. Claims And Evidence (L78-82)
> Our overall objective is to have a fully unsupervised and automated pipeline for designing foundational models for DSR. The several heuristic methods we propose in our gating mechanism are designed to be quick and interpretable. We do note, however, that other heuristics could be considered, and we plan to do so in future work (see Q3 to Reviewer rCHt).
>
> We also note that manual partitioning can be detrimental. For instance, Figure 6 shows how vanilla K-Means (first column) provides a less clear distinction between the classes compared to MixER with 3 experts.
>
> Finally, adaptation with MixER in Section 4.1 reveals similarity to meta-training. While we did not include timings, adaptation with the backbone meta-learners is relatively quick compared to the meta-training stage.
>
> ---
> ### Q2. Claims And Evidence (L100-104)
>
> Indeed, our claim from L100-104 is mostly based on Section 2.2, and we apologize for not explicitly stating the number of experts used (which is $M=2$ as seen in Figure 8). That said, we respectfully disagree with the reviewer's comment invalidating our experimental protocol. Since the hierarchy in the data is unknown, load balancing terms are not suitable (see Q3). The hyperparameters (mentioned in Appendix C.3) are the same for naive MoE and MixER (see Q5 to Reviewer rCHt). We believe this makes the comparison in Figure 2 fair and supportive of our claim.
> Finally, we note that the limitations of gradient descent as a routing mechanism persist when the number of experts is scaled up (see W2 to Reviewer Cntw).
>
>
> ---
> ### Q3. Claims And Evidence (L184-185)
> The clusters can be of unequal size. We do not enforce an equal split of environments, nor do we necessarily want such a split. Indeed, if the families fundamentally possess different numbers of environments, we want the model to automatically pick up on this, and specialize appropriately.
>
> In summary, although Figure 8 suggests that the clusters are equally sized (as they should be as per Table 1 with ODEBench), we recommend using Figure 4, which provides a more holistic overview of our approach.
>
> ---
> ### Q4. Experimental Design: Parameter Count Issues?
>
> In section 4.2, we mention that the compared methods have the same **active** number of parameters (see Table 4, repeated in L294-295). In other sections, we retain the same active parameter count. We will add this information in the Introduction to Section 4.
>
> ---
> ### Q5. Experimental Design: Overfitting?
>
> We ensure fair comparison by picking the best model based on the validation loss (Figure 2, bottom). The reason why naive MoE and task-specific meta-learning fail is very clearly due to inconsistent routing, as we show with Figure 8. If indeed, overfitting was occurring, we wouldn't observe the excellent Relative L2 error on the left-out test set in Figure 2 (top).
>
> ---
> ### Q6. Experimental Design: Train-Set MSE ?
>
> We agree that "Train MSE" is not the most intuitive term for this column. This is indeed not the error on the train set. Instead, it is the error on the meta-training's query set. Similarly, the "Adapt" column indicates the error on the adaptation's query set. These splits are explained in Table 5.
>
> ---
> ### Q7. Quality of the Code
>
> We have updated our code. Hopefully, this improved codebase can prompt the reviewer to comment on the correctness of our implementation.
>
> ---
> ### Q8. Other Strengths And Weaknesses
>
> We respectfully disagree with the reviewer that no significant strengths are successfully demonstrated. As per our reply in Q2+Q5 above, we believe Figure 2 vividly demonstrates the value of MixER. Furthermore, Figure 6 and Q4 to Reviewer 6AnX strengthen our claim.
>
> The weaknesses of our approach are analyzed and fully documented, and we believe knowledge of those limitations is valuable for the community moving forward.
>
> ---
> ### Q9. Figure 4 Equal Split Typo?
>
> There is no typo. Figure 4 shows two clusters of different sizes (indeed, 3 and 5), and the derived $Y$ labels have a corresponding 3:5 split. As per our response in Q3, there is no procedure to ensure an equal split (which isn't necessarily desirable).
>
> ---
> ### Q10. Additional References
>
> We are grateful to the reviewer for these references, and we will make sure to include them in the final version of the paper.

---

> > ### Comment · Reviewer_p7Cj · 2025-04-08
> >
> > Thank you for taking time to respond to my review.
> >
> > While the clarifications are appreciated, I remain concerned about the experimental protocol's rigour. Matching the active number of parameters doesn't ensure a fair comparison, as the proposed MoE models have M times more total weights. This not only gives them greater representational power compared with non-MoE models, but also makes them more prone to overfitting. Consequently, a proper evaluation requires comparison against (at least) carefully tuned MoE baselines.
> >
> > Taking this into account, as well as other reviews, I think the promising results shown in Figure 2 are not sufficient on their own to convincingly demonstrate the proposed model's viability and generality, so I keep my current score.

---

> > > ### Author Response · Authors · 2025-04-09
> > >
> > > Dear reviewer, thank you for acknowledging our rebuttal and for providing these additional comments.

---

### Official Review · Reviewer_rCHt · 2025-03-15

**Overall Recommendation:** 1

**Summary:**

This paper addresses the challenge of dynamical system reconstruction (DSR) in data-scarce environments, where traditional meta-learning approaches struggle to generalize across loosely related system hierarchies. To overcome these limitations, the authors propose MixER (Mixture of Expert Reconstructors), a sparse Mixture of Experts (MoE) framework that replaces conventional gradient-based gating with a K-means clustering and least squares optimization mechanism. Such routing strategy enables more efficient expert specialization and adaptation, facilitating hierarchical meta-learning across diverse dynamical systems. Extensive experiments demonstrate that MixER achieves superior efficiency and scalability in low-data, heterogeneous settings, successfully reconstructing systems governed by up to ten parametric ordinary differential equations (ODEs). However, its performance deteriorates in high-data regimes, particularly when experts are constrained to process highly correlated datasets, limiting their ability to generate meaningful contextual representations.

**Claims And Evidence:**

Following claims are well-supported.

- The experimental comparisons convincingly demonstrate the limitations of existing methods in handling hierarchical DSR.

- The experiments clearly illustrate how the new routing mechanism improves expert allocation.

However,  I think the performance of MixER is highly unstable. For example, in Table 2, it cannot outperform naïve MoE w.r.t. train MSE with GEPS backbone, and adapt MSE with CoDA, GEPS backbone. Moreover, it cannot outperform MixER with only 1 expert, as shown in Table 2 and 3. Why could that happen? Also, while the experiments confirm improved generalization across loosely related environments, the drop in performance on highly related datasets contradicts the claim of universal generalization.

**Essential References Not Discussed:**

There is no essential reference which is not discussed.

**Experimental Designs Or Analyses:**

Also see Claims and Evidence for more information.

1. Few-Shot Learning on ODEBench-10A and ODEBench-10B:

a. The paper do not provide any other baseline results.

b. The adaptation phase uses only one environment for training (Table 5), which may not reflect real-world few-shot scenarios requiring multiple support samples.

c. The lack of discussion on hyperparameter tuning (e.g., K-means iterations, learning rates) raises questions about reproducibility and sensitivity.

2. Time Series Classification on Epilepsy2

a. Without full results, it’s unclear how MixER handles the test set imbalance—accuracy alone may be misleading without precision/recall.

b. The single-channel EEG focus limits complexity compared to multi-channel data, potentially overstating performance.

c. Batch size (571) during adaptation (Appendix C.3) seems unusually large for few-shot learning, requiring clarification.

**Methods And Evaluation Criteria:**

The proposed MixER method is a sparse top-1 MoE layer with K-means and least square-based gating mechanism. It seems to be suited for hierarchical dynamical system reconstruction in sparse, multi-family datasets. It has potential to address limitations of traditional meta-learning by enabling simultaneous learning across loosely related systems while retaining adaptability, making it apt for scientific applications like neuroscience. However, its top-1 architecture may limit performance in high-data regimes.

Evaluation criteria, including relative L2 error, MSE, and classification accuracy, appropriately assess reconstruction and generalization. Benchmarks like ODEBench, classical DSR datasets (e.g., Lotka-Volterra), and Epilepsy2 EEG data cover diverse scenarios, aligning with the problem’s scope.

**Other Comments Or Suggestions:**

N/A

**Other Strengths And Weaknesses:**

Strengths

1. The paper proposes a new gating mechanism which can potential benefit the research of MoE models.

2. The paper is relatively well-written and easy to understand.

3. The focus on low-data regimes aligns with real-world scientific challenges (e.g., neuroscience, clinical data), making it a valuable contribution to foundational models for scientific discovery.

Weaknesses

1. Unclear Motivation for MoE: The rationale for choosing MoE over other architectures (e.g., single large networks, transformers) is not well-articulated. While inspired by large language models, the paper lacks a deeper justification linking MoE’s sparsity to DSR’s unique demands.

2. Insufficient Baselines: The baseline selection is incomplete. It omits comparisons with non-meta-learning methods or other scalable architectures (e.g., Mamba,) limiting the scope of validation.

3. Neglect of Alternative Routing Methods: The paper focuses solely on K-means and least squares routing without exploring other strategies. This narrows the methodological exploration and weakens claims of optimality.

4. Limited Theoretical Rigor: Theoretical claims (e.g., gating superiority, optimization convergence) lack formal proofs, relying heavily on empirical results, which reduces their universality and robustness.

5. Reproducibility Concerns: Key hyperparameters (e.g., learning rates, K-means initialization details) are underspecified, potentially hindering replication and sensitivity analysis.

**Questions For Authors:**

My questions are in lines with above concerns, issues and weakness part.

**Relation To Broader Scientific Literature:**

The key contributions of "Towards Foundational Models for Dynamical System Reconstruction: Hierarchical Meta-Learning via Mixture of Experts" align with and extend several threads in the broader scientific literature on dynamical system reconstruction (DSR) and meta-learning.

1. Hierarchical Meta-Learning for DSR

2. Mixture of Experts (MoE) in Scientific Modeling

3. Foundational Models in Science

**Theoretical Claims:**

N/A. There is no proofs in this manuscript.

---

> ### Author Rebuttal · Authors · 2025-03-31
>
> We sincerely thank the reviewer for reading and commenting on our work. We are happy you praised the novelty of our gating mechanism from which MoE research stands to benefit. Your appreciation of how well our paper was written and how our exposition made it easy to understand is encouraging. The fact that you found our contribution valuable to foundational models for scientific discovery is also encouraging. Indeed, our end goal is to solve real-world scientific challenges, especially in the low-data regime in which foundational models tend to be fine-tuned (e.g., neuroscience, clinical data).
>
> ---
> ### Q1. Unclear Motivation for MoE
>
> DSR is unique in that its datasets are very different from one source to another, contrary to language, which inherently contains lots of structure. While large monolithic models are powerful, they do not explicitely provide specialization across distinct behaviors. Another argument for using MoEs for DSRs is that they are computationally efficient since only a small subset of experts is active, not to mention increased interpretability.
>
> The limitations of single monolithic models have been explored in [1,2]. As for transformers meta-learning DSRs, we can refer to [3].
>
> Finally, as our proposal is simply an augmentation of existing meta-learners, it does not prohibit the use of underlying transformer backbones. While we do not pursue this avenue, we believe the community will find it helpful.
>
> To clarify this motivation, we will expand our introductory paragraph in the paper to indicate that our choice of MoE is driven by more than the success of recent large language models.
>
> #### References:
> [1] Yin et al. LEADS: Learning dynamical systems that generalize across environments, NeurIPS, 2021.
> [2] Nzoyem et al., Neural Context Flows for Meta-Learning of Dynamical Systems, ICLR 2025
> [3] Serrano et al., Zebra: In-Context and Generative Pretraining for Solving Parametric PDEs, arXiv, 2024.
>
> ---
> ### Q2. Insufficient Baselines
>
> Our objective with this work is to show that preexisting **meta-learners** can be augmented with MixER. As such, we believe it is important to show results with and without MixER, as we've done repeatedly. As for non-meta-learning (which isn't the focus of our work), please see Q1.
>
> Concerning the use of other scalable architectures like Mamba, we'd be happy to incorporate any work that specializes it for non-linear dynamical systems across parameter changes into our MixER formulation.
>
> ---
> ### Q3. Neglect of Alternative Routing Methods
>
> We selected K-means and least squares routing as they provide a **simple** yet effective unsupervised clustering mechanism for expert assignment. However, we acknowledge that other routing strategies (e.g., soft gating, reinforcement learning-based routing) could further improve performance.
>
> Our goal was to keep the routing method **interpretable** and computationally **efficient**. That said, we deeply appreciate the suggestion to explore alternative routing mechanisms and will investigate this in future work.
>
> ---
> ### Q4. Limited Theoretical Rigor
>
> Indeed, our approach primarily relies on empirical validation as MixER routing for DSR is inherently data-dependent and difficult to formalize in a universal manner. It would take significant efforts to provide such guarantees, perhaps even a full paper. We will add this theoretical limitation in Section 6.
>
> ---
> ### Q5. Reproducibility Concerns
>
> While we included major hyperparameters in Appendix C.3, we recognize that details such as K-means initialization (see Algorithm 2, line 4) could be more thoroughly documented. Our code partly addresses this problem, with all hyperparameters clearly stated at the start of each `main_mixer.py` script in the `./examples` folder. Due to rebuttal limitations, we cannot write down more details here.
>
> ---
> ### Q6. Claims And Evidence: Why could that happen?
>
> The one powerful expert meta-learner is exposed to more shared knowledge than any of the 10 or 20 experts in Tables 2 and 3. Please see Q7+Q5 from Reviewer 6AnX and Q4+W1 from Reviewer Cntw for more on this. Keeping this limitation in mind, we do not claim universal generalization.
>
> ---
> ### Q7. Experimental Design Or Analyses
> - 1b) We agree that this may not reflect real-world scenarios, but this is the meta-learning for DSR setting inherited from [1,2]. We had little control over this.
> - 2c) The adaptation batch size of 571 (Appendix C.3) was chosen for computational efficiency. With 11,420 adaptation environments, it would have been intractable to adapt one at a time. We split them into 20 batches, resulting in a batch size of 571.
>
> ---
> We could not respond to all the reviewer's concerns due to character limitations (especially in Q7), and we are happy to answer any additional concern. We thank you once more for your comments, as they've already helped improve our paper.

---

### Official Review · Reviewer_Cntw · 2025-03-18

**Overall Recommendation:** 2

**Summary:**

The authors propose MixER, an Mixture-of-Experts routing mechanism for learning diverse dynamical systems while preserving adaptability within individual environments. It incorporates an environment-specific context vector for better gating decisions, and integrates k-means clustering for hierarchical learning. While well-motivated, the method lacks strong experimental evidence of its effectiveness, and some details need further clarification.

**Claims And Evidence:**

Some of the claims are well-supported, others need further clarification. For instance,“We identify a fundamental limitation of gradient descent when routing contextual information to DSR models, which slows down expert specialization when training MoEs.” --> Other than the example in Section 2.2, is there other evidence for this claim?

**Essential References Not Discussed:**

N/A

**Experimental Designs Or Analyses:**

The experimental designs are mostly standard, but I would appreciate more meaningful model comparisons and thorough ablation studies.

**Methods And Evaluation Criteria:**

Yes, the methods and evaluation criteria are mostly valid and appropriate.

**Other Comments Or Suggestions:**

Figure 4 is difficult to understand, please improve clarity and captioning.

**Other Strengths And Weaknesses:**

Strengths: Using a context vector as a gating input is a reasonable and interesting approach. The authors also conducted extensive experiments across multiple datasets.

Weaknesses: The empirical results are not compelling and do not support the claim well.

**Questions For Authors:**

Can this model be extended into a foundation model for ODEs? What potential aspects can be further explored? Can this cross-family learning simply be cast into a multi-task learning problem?

The results in Fig. 2 do not appear to align with the trends observed in Tables 2 and 3, where MixER-10 does not clearly outperform other baselines. Could you clarify the underlying reason?

**Relation To Broader Scientific Literature:**

This article sits in between a few research directions including Neural ODE, scientific foundation models, multitask learning,  meta-learning, etc. It redefines a research topic of learning across diverse dynamical systems into meta-learning and proposes a specific methodology to this formulation.

**Theoretical Claims:**

N/A

---

> ### Author Rebuttal · Authors · 2025-03-30
>
> We thank the reviewer for taking the time to read and provide comments on our paper. We are happy you found the approach well-motivated and interesting while acknowledging how we redefined a research topic of learning across diverse dynamical systems into meta-learning. We are pleased you found our results extensive across multiple datasets.
>
> ---
> ### Q1. Can this model be extended into a foundation model for ODEs?
> Our model can indeed be extended into a foundational model for ODEs, especially since ODEs come in various types (bifurcation, time scales, etc.). It is difficult to imagine a foundational model for DSR without involving a mixture of specialized experts.
>
> We view hierarchical meta-learning as a possible route toward cheap foundational models for ODEs. We envision several issues along the way, the main one being the diminished representational power when each expert is exposed to only a fraction of a very abundant dataset (see Q4 from Reviewer 6AnX). We point this out in our limitations, and it constitutes a solid **warning** to use varied datasets for training foundational DSR models.
>
> ---
> ### Q2. What potential aspects can be further explored?
> Our results highlight the important promises of this approach. As mentioned above, future work will explore ways to retain the capabilities of task-specific meta-learners within MixER. Specifically, we will investigate when MixER converges to only using the minimum number of required experts to fully represent the data, for optimal performance. Other important areas of future work were pointed out in our responses to Reviewer 6AnX.
>
> ---
> ### Q3. Can this cross-family learning simply be cast into a multi-task learning problem?
> We agree that this cross-family learning can be cast into a multi-task learning problem. However, to maintain the ability of these models to quickly adapt—to be **easily** fine-tunable for downstream tasks (like we believe foundational models should be)—we choose to incorporate meta-learning, thus building on rapidly growing research in this area.
>
> ---
> ### Q4. Could you clarify the underlying reason why MixER doesn't consistently outperform the baselines?
> This is because the dataset used in Figure 2 is much sparser than the ones for Table 2 and Table 3 (as we discuss in Table 1 and the Appendix). In Figure 2, the families are clearly different, and there is very little room for ambiguity in the routing; while in Figure 5 (related to Table 3), we have 10 families, and trajectories in some families (in chunks of 16) are similar to trajectories in other families. When compared to Figure 8 (related to Figure 2), we see that routing is not clean anymore. This routing uncertainty is at the source of the performance drop.
>
> All in all, with abundant data and little family discrepancy, MixER struggles to outperform the task-specific meta-learner, which can more effectively extract shared knowledge from the whole dataset.
>
> ---
> ### W1. Weak experimental evidence
> We acknowledge that our presentation of experimental evidence in the main Introduction was lacking and does not promote our approach very well. We believe, however, that a better presentation as follows is possible:
> - Exceptional results on ODEBench-2 (starkly different datasets) and on SCCTS
> - Underperformance on ODEBench-10A and on Epilepsy2
>
> The common denominator is that MixER performs well when little data is available for very different environments.
>
> ---
> ### W2. Additional evidence of the weakness of GD ?
> No, we do not include any other additional evidence for this. In the manuscript, Section 2.2 is the only experiment specifically designed to illustrate the limitations of gradient descent.
>
> We note, however, that the poor routing was observed throughout all our experiments (e.g. https://anonymous.4open.science/r/MixER/assets/gate_values_geps_gd_10M.pdf). We will include similar plots in the final manuscript to indicate that the competitive performance of Naive MoE on some problems is not due to clever utilization of its experts, but to strong generalizability of one or a few of its base meta-learners.
>
> ---
> ### W3. More results
> We appreciate your request for more meaningful results and ablation studies. But seeing as our paper is essentially a giant ablation study of gradient descent as the routing mechanism, we do not see which other ablation can be performed. Perhaps the reviewer could suggest a few datasets and specific experiments?
>
> ---
> ### W4. Figure 4 captioning
> Thank you for this suggestion: we propose to add to the arrows the label numbers (1), (2), (3), or (4), and change the caption to "_Illustration of the main stages of our context-based gating update algorithm: (1) K-means clustering; (2) per-expert per-environment loss computation; (3) expert-cluster pairings; and (4) least-squares optimization. All tensors are in compatible shapes._"
>
> ---
> Once again, thank you for your positive and valuable feedback. Your insights have helped us improve the clarity of our paper.

---

> > ### Comment · Reviewer_Cntw · 2025-04-09
> >
> > I would like to thank the authors for their response.
> >
> > >Response to Q4
> >
> > There should be a remedy for this performance drop when families are similar. If adding more data causes problems in model performance, I do not see adequate scaling ability of the model and efficacy of the methodology. And in principle, when families are similar, I would expect the model to learn even better.
> >
> > This would be a critical point to resolve in order to make this article convincing and significant enough to be accepted.

---

### Decision · Program_Chairs · 2025-05-01

**Decision:**

Reject

**Comment:**

This work discusses a meta-learning approach for dynamical systems reconstruction based on a variation of the mixture-of-experts architecture. With an eye on foundation models for dynamical systems, the paper addresses a timely and important topic, and follows a promising new direction. However, essentially all four referees saw the experimental evaluation and support w.r.t. the stated claims and goals as somewhat deficient or weak, with the authors’ method not convincingly and consistently outperforming baselines. They also brought up various concerns relating to the methodology and tested baselines, and the lack of a theoretical underpinning.

While the authors addressed some of these shortcomings in their rebuttal, overall the referees remained rather unconvinced.